# WASSERSTEIN HYPERGRAPH NEURAL NETWORK

## ABSTRACT

The ability to model relational information using machine learning has driven advancements across various domains, from medicine to social science. While graph representation learning has become mainstream over the past decade, representing higher-order relationships through hypergraphs is rapidly gaining momentum. In the last few years, numerous hypergraph neural networks have emerged, most of them falling under a two-stage, set-based framework. The messages are sent from nodes to edges and then from edges to nodes. However, most of the advancement still takes inspiration from the graph counterpart, often simplifying the aggregations to basic pooling operations. In this paper, we are introducing Wasserstein Hypergraph Neural Network, a model that treats the nodes and hyperedge neighbourhood as distributions and aggregates the information using Sliced Wasserstein Pooling. Unlike conventional aggregators such as mean or sum, which only capture first-order statistics, our approach has the ability to preserve geometric properties like the shape and spread of distributions. This enables the learned embeddings to reflect how easily one hyperedge distribution can be transformed into another, following principles of optimal transport. Experimental results demonstrate that applying Wasserstein pooling in a hypergraph setting significantly benefits node classification tasks, achieving top performance on several real-world datasets.

## 1 INTRODUCTION

The potential to learn from relational data has substantially broadened the applicability of machine learning, extending its reach to a wide range of fields (Johnson et al., 2023; Tong et al., 2022; Sanchez-Gonzalez et al., 2020; Lam et al., 2023; Monti et al., 2019; Gilmer et al., 2017; Huang et al., 2020). The flexibility of graph structures makes them well-suited for representing natural phenomena involving various types of interactions. However, while graphs are restricted to model pairwise connections, many real-world interactions involve more than two entities. To fill this gap, a generalisation of graphs called hypergraphs was introduced, allowing the representation of relationships among multiple elements.

More precisely, a hypergraph is characterised by a set of edges, where each edge connects a set of nodes, potentially of varying cardinality. The challenge of designing hypergraph networks becomes the challenge of properly modelling these sets. Many approaches (Chien et al., 2022; Wang et al., 2022; Huang & Yang, 2021a) tackle this using a two-step process: first, the model aggregates information from the nodes within each hyperedge to compute a representation for that hyperedge. Then, it updates each node's representation using information from the hyperedges it belongs to. Both steps rely on methods designed to handle sets of elements.

Although set representation learning has seen significant progress in recent years (Xie & Tong, 2025), hypergraph networks still largely rely on sum-based aggregation methods such as Deep Sets (Zaheer et al., 2017) and Set Transformers (Lee et al., 2019). Table 1 presents the update rules of several widely used hypergraph networks, emphasising that each of them utilises a form of the sum-based aggregator. Despite their strong theoretical foundation, these aggregators can struggle to effectively capture the full geometry of set-structured inputs (Naderializadeh et al., 2021).

In this work, we introduce Wasserstein Hypergraph Neural Networks (WHNN), a model that uses Sliced Wasserstein Pooling (SWP) (Naderializadeh et al., 2021) as node and hyperedge aggregator. This pooling is based on the Wasserstein distance, an optimal transport metric which measures the distance between two distributions based on the cost of transporting mass from one to another. We

Table 1: **Overview of the update rules used as aggregation steps** in various hypergraph neural networks from the literature. While theoretically powerful, summing can easily destroy all the geometric relationships between points. $\mathcal{N}_v(i)$ is the neighbourhood of node $v_i$ of cardinality $d_i$, $\mathcal{N}_e(j)$ is the neighbourhood of edge $e_j$ of cardinality $d_j$ and $\epsilon$, $W_*$, $\tilde{W}_*$ are learnable parameters.[1]

| Model | Hyperedge aggregation | Node aggregation |
|---|---|---|
| HGNN | $h_e \leftarrow \sum_{i \in \mathcal{N}_e(e)} \frac{1}{\sqrt{d_i}} x_i W$ | $x_i \leftarrow \frac{1}{\sqrt{d_i}} \sum_{e \in \mathcal{N}_v(i)} \frac{1}{d_e} h_e$ |
| HCHA | $h_e \leftarrow \sum_{i \in \mathcal{N}_e(e)} \alpha_{e,i} x_i W$ | $x_i \leftarrow \sum_{e \in \mathcal{N}_v(i)} \tilde{\alpha}_{i,e} h_e \tilde{W}$ |
| UniGIN | $h_e \leftarrow \sum_{i \in \mathcal{N}_e(e)} x_i$ | $x_i \leftarrow \sum_{e \in \mathcal{N}_v(i)} h_e W + (1 + \epsilon) x_i W$ |
| ED-HNN | $h_e \leftarrow \sum_{i \in \mathcal{N}_e(e)} \mathrm{MLP}(x_i)$ | $x_i \leftarrow \sum_{e \in \mathcal{N}_v(i)} \mathrm{MLP}(x_i \| h_e)$ |
| AllDeepSets | $h_e \leftarrow \mathrm{MLP}(\sum_{i \in \mathcal{N}_e(e)} \mathrm{MLP}(x_i))$ | $x_i \leftarrow \mathrm{MLP}(\sum_{e \in \mathcal{N}_v(i)} \mathrm{MLP}(h_e))$ |
| AllSetTransformer | $h_e \leftarrow \sigma(\sum_{i \in \mathcal{N}_e(e)} (\alpha_i x_i W_v)$ | $x_i \leftarrow \sigma(\sum_{e \in \mathcal{N}_v(i)} (\tilde{\alpha}_e h_e \tilde{W}_v)$ |

argue that this geometric information is highly relevant for hypergraph learning. Our experimental results support this claim, showing that WHNN not only outperforms traditional sum-based aggregation methods used in previous hypergraph models but also achieves superior performance compared to several strong hypergraph methods across a range of real-world datasets for node classification.

**Our main contributions** are summarised as follow:

1. We propose **a novel hypergraph architecture, Wasserstein Hypergraph Neural Network (WHNN)**, which leverages Sliced Wasserstein Pooling for both node and hyperedge aggregation to more effectively capture the geometric structure of the feature space.

2. We empirically show that Wasserstein aggregation is highly effective for hypergraph representation, consistently **outperforming traditional sum-based methods** such as Deep Sets and Set Transformers, regardless of the encoder used to process the nodes.

3. Wasserstein Hypergraph Network achieves top results on multiple real-world datasets for node classification, highlighting the advantages of incorporating optimal transport into hypergraph processing.

## 2 RELATED WORK

**Hypergraph representation learning.** Hypergraphs represent a versatile structure for modelling group-wise interactions, which allows us to capture interactions between various numbers of elements. This flexibility, combined with the widespread presence of higher-order interactions in real-world scenarios, has led to a growing interest in developing machine learning architectures for modelling hypergraph data. Some methods (Feng et al., 2019; Tang et al., 2024) reduce the hypergraph to a clique-expansion graph that can be further processed with standard graph neural networks. A more popular approach is based on a two-stage framework (Chien et al., 2022; Huang & Yang, 2021a), which sends the information from node to hyperedges and then from hyperedges back to nodes. Depending on how these stages are instantiated, several architectures emerged. HCHA (Bai et al., 2021) and HERALD (Zhang et al., 2022) use an attention mechanism to combine the information. AllDeepSets (Chien et al., 2022) uses Deep Set model (Zaheer et al., 2017), while AllSetTransformer (Chien et al., 2022) uses a PMA-like (Lee et al., 2019) pooling.

In all of these methods, the information sent from the node is independent of the target hyperedge. Recently, models that create edge-dependent node representations have gained traction. ED-HNN (Wang et al., 2022) uses as messages a concatenation of node and hyperedge information, while MultiSet-Mixer (Telyatnikov et al., 2025) uses MLP-Mixer (Tolstikhin et al., 2021) to combine the information. Similar to our node encoder, CoNHD(Zheng & Worring, 2025) incorporates pairwise propagation at the hyperedge-level using self-attention blocks (Lee et al., 2019) to create edge-dependent representations. However, similar to Choe et al. (2023), the model is only tested on hyperedge-dependent node classification tasks, where each node is assigned multiple labels corresponding to the number of

---

[1]The coefficients $\alpha_{e,i}$ used in summations are scalars predicted as $\mathrm{MLP}(x_i \| h_e)$ and $\sigma$ is a composition of skip connections and layer normalisation, while $\alpha_i = (\theta W_q)(x_i W_k)^T$ with $\theta$, $W_q$ and $W_k$ as parameters.

hyperedges it participates in. A complementary line of work (Wang et al., 2024) represents uniform hypergraphs as high-dimensional tensors and applies tensorial operators to propagate the information.

In contrast, we are interpreting the hyperedges as samples from a set of probability distributions, and use Sliced Wasserstein Pooling to aggregate the information such that we preserve geometric information. In terms of node encoders, we are experimenting with both edge-dependent and edge-independent modules.

**Set representation learning.** The core operation in set representation learning is the permutation-invariant operator that aggregates the information without imposing an order among elements. Popular examples of such operators include summation, mean or maximum. More recently, learnable versions of permutation-invariant poolings were introduced. Among these, Deep Sets (Zaheer et al., 2017) uses element-wise encoding of the elements followed by summation and is proven to be a universal approximator for permutation-invariant functions. Janossy Pooling (Murphy et al., 2019) extends this model by explicitly aggregating pairs of elements. On the other hand, Set Transformer (Lee et al., 2019) and RepSet (Skianis et al., 2019) use an anchor set as a reference and compute the similarity against this set as a representation, while FSPool (Zhang et al., 2019) sorts the elements feature-wise to create a canonical order. Recently, Kothapalli et al. (2024) shows empirically that combining an equivariant backbone with an invariant pooling layer creates powerful set representation learning. Inspired by optimal transport literature, Sliced Wasserstein Pooling was introduced in Naderializadeh et al. (2021) as a geometrically-interpretable set representation technique.

**Wasserstein embeddings.** In recent years, Wasserstein distance has attracted significant attention in deep learning, demonstrating success in areas such as generative modeling (Arjovsky et al., 2017; Nguyen et al., 2021a), natural language processing (Frogner et al., 2019) and point cloud processing (Nguyen et al., 2021b). In graph representation learning, Wasserstein barycenters (Chen et al., 2025) were applied to model uncertainty in graphs with missing attributes. On the other hand, Wasserstein distance was used to define a similarity kernel between pairs of graphs (Togninalli et al., 2019). While recognised as a powerful tool, computing this distance for each pair of compared graphs is extremely inefficient. More recent works (Kolouri et al., 2021; Mialon et al., 2021; Courty et al., 2018) try to reduce this cost by introducing Wasserstein embeddings. The purpose of a Wasserstein embedding is to infer a vector representation such that the $L_2$ distance in the vector space approximates the Wasserstein distance in the input space. Particularly important for us is the work of Naderializadeh et al. (2021) which produces set representations using efficient Wasserstein embeddings.

In order to more effectively capture the internal structure of node and hyperedge neighbourhoods, in this work we employ Sliced Wasserstein Pooling as the aggregation operator in hypergraph message passing, demonstrating its advantages for hypergraph representation learning. Note that developing strong and efficient embeddings that preserve the Wasserstein distance is an active area of study. Recent advances in Wasserstein embeddings (Amir & Dym, 2025) provide stronger theoretical guarantees than Sliced Wasserstein Pooling and could potentially be leveraged to further enhance our approach in future work.

## 3 BACKGROUND

### 3.1 HYPERGRAPH REPRESENTATION LEARNING

A hypergraph is a tuple $\mathcal{H} = (V, E)$ where $V = \{v_1, v_2 \dots v_N\}$ is a set of nodes, and $E = \{e_1, e_2 \dots e_M\}$ is a set of hyperedges. Different from the graph structure, where each edge contains exactly two nodes, in a hypergraph, each hyperedge contains a set of nodes, which can vary in cardinality. Each node $v_i$ is characterised by a feature vector $x_i \in \mathbb{R}^d$. We denote by *neighbourhood of hyperedge* $e_i$ the set of nodes that are part of that hyperedge $\{v_j | v_j \in e_i\}$. Similarly, the *neighbourhood of a node* $v_i$ is the set of all hyperedges containing that node $\mathcal{N}_{v_i} = \{e_j | v_i \in e_j\}$.

Several architectures were developed for hypergraph-structured input (Feng et al., 2019; Wang et al., 2022; Huang & Yang, 2021b; Chien et al., 2022). However, the most general pipeline follows a two-stage framework, inspired by the bipartite representation of the hypergraphs. First, the information is sent from nodes to the hyperedges using a permutation-invariant operator $z_j = f_{V \to E}(\{x_i | v_i \in e_j\})$. Secondly, the messages are sent back from hyperedge to nodes $\tilde{x}_i = f_{E \to V}(\{z_j | v_i \in e_j\})$.

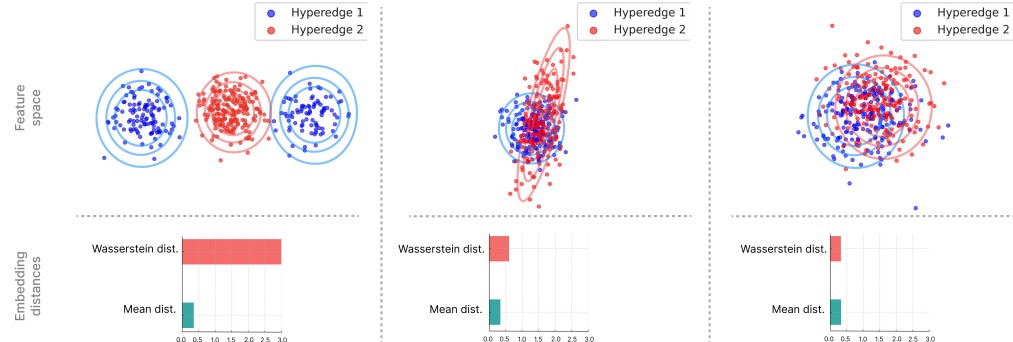

Figure 1: **Comparison between Wasserstein and average aggregators.** For each pair of synthetic hyperedges, the *Wasserstein distance* is approximated by the euclidean distance between the Sliced Wasserstein embeddings as computed by our WHNN model, whereas the *Mean distance* is computed as the euclidean distance between their mean-pooled representations. While mean aggregators fail to differentiate among the three scenarios, Wasserstein aggregators captures these geometric differences.

While aggregators like Deep Sets (Zaheer et al., 2017) were theoretically capable of approximating any permutation-invariant function on sets, they rely on the initial encoder (such as MLPs) to reshape the feature space in a way in which the sum pooling does not lose important information. Even if such a feature space exists in theory, finding an appropriate projection in practice can be challenging for certain applications (Kothapalli et al., 2024), limiting the effectiveness of simple sum-based aggregation.

In this work, we are following the standard two-stage framework. Compared to existing methods, we take advantage of the success demonstrated by Sliced Wasserstein Pooling in capturing and retaining the geometric structure of sets and propose the first hypergraph model that uses optimal transport techniques to perform the node and hyperedge aggregation.

### 3.2 SLICED WASSERSTEIN POOLING (SWP)

To ensure the method's readability, this section introduces all the key concepts underlying our Wasserstein Hypergraph Neural Network. First, we will define the 2-Wasserstein metric, approximate it using the tractable Sliced-Wasserstein distance and finally present the algorithm to compute the SWP used as an aggregator in our model.

**Definition 1.** The **2-Wasserstein distance** between two distributions $p_i$ and $p_j$ over $\mathbb{R}^d$ is defined as:

$$\mathcal{W}_2(p_i, p_j) = \left( \inf_{\gamma \in \Gamma(p_i, p_j)} \int_{\mathbb{R}^n \times \mathbb{R}^n} ||x - y||^2 d\gamma(x, y) \right)^{\frac{1}{2}}, \tag{1}$$

where $\Gamma(p_i, p_j)$ represent the collection of all the transport plans with marginals $p_i$ and $p_j$.

In simpler terms, the 2-Wasserstein distance quantifies the cost of transforming one distribution into another. Unfortunately, computing the infimum over all possible transport maps is generally untractable. However, **in the one-dimensional case (when $d = 1$ ), a closed-form solution exists** that avoids expensive optimisation. Specifically, when $p_i$ and $p_j$ are probability distributions over $\mathbb{R}$, the 2-Wasserstein distance is given by $\mathcal{W}_2(p_i, p_j) = \left( \int_0^1 |F_{p_i}^{-1}(t) - F_{p_j}^{-1}(t)|^2 dt \right)^{\frac{1}{2}}$, where $F_{p_i}^{-1}$ and $F_{p_j}^{-1}$ denote the inverse cumulative distribution functions of $p_i$ and $p_j$. A key practical benefit of this formulation is that this integral can be empirically estimated using a discrete sum over sorted samples from the distribution.

Building on this observation, Sliced Wasserstein distance (Bonneel et al., 2015) was introduced to approximate the Wasserstein distance, by projecting the high-dimensional probabilities into 1D lines using all possible directions on the unit sphere.

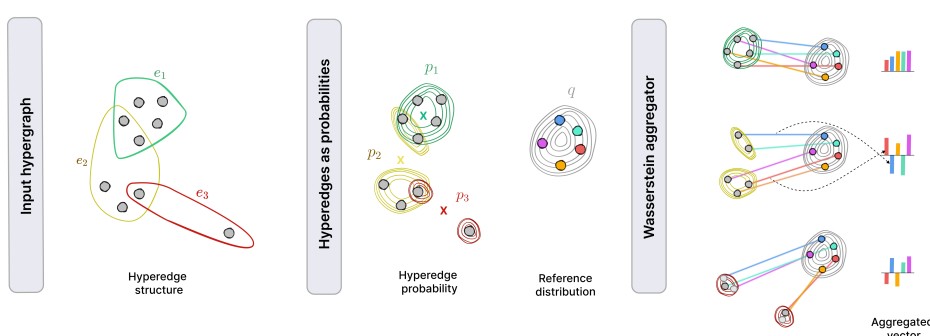

Figure 2: **One stage (node-to-hyperedge) of WHNN pipeline** designed to better reflect the geometric relationships among nodes features in the hyperedge compared to the traditional aggregators. The hypergraphs are viewed as a collection of probability distributions $\{p_i\}$, one for each hyperedge, with the observed nodes treated as samples drawn from it. An additional distribution $q$ is picked as a reference. Finally, **Sliced Wasserstein Pooling** is used as the aggregation method. Each hyperedge is represented by its Wasserstein distance to a reference distribution: for every hyperedge, the distances between its sorted elements and the reference samples are computed (depicted as colored arrows). The length of each arrow forms an entry in the final vector representation (shown as colored bars).

**Definition 2.** The **Sliced Wasserstein distance** between two distributions $p_i$ and $p_j$ over $\mathbb{R}^d$ is defined as:

$$\mathcal{SW}_2(p_i, p_j) = \Big( \int_{S^{d-1}} \mathcal{W}_2(P_\theta p_i, P_\theta p_j) d\theta \Big)^{\frac{1}{2}} \approx \Big( \frac{1}{L} \sum_{l=1}^{L} \underbrace{\mathcal{W}_2(P_{\theta_l} p_i, P_{\theta_l} p_j)}_{\text{1D Wasserstein distance}} \Big)^{\frac{1}{2}}, \qquad (2)$$

where $S^{d-1}$ is the unit sphere in $\mathbb{R}^d$, $P_\theta p_i$ is the projection (pushforward) of $p_i$ onto the line direction $\theta$ and $\{\theta_l\}_{l=1}^{L}$ represents the set of $L$ directions used to empirically approximate the expectation.

To avoid the computational cost of calculating distances between every pair of probability distributions, the **Sliced Wasserstein embedding** (Naderializadeh et al., 2021) was proposed. It maps a probability distributions $p_i$ to a vector $\phi(p_i)$ in such a way that the Euclidean distance between the vectors (which is inexpensive to compute) approximates the Sliced Wasserstein distance between the original distributions $||\phi(p_i) - \phi(p_j)||_2 \approx \mathcal{SW}_2(p_i, p_j)$. In other words, it provides a vectorial representation that captures the geometric structure of distributions, preserving information about how costly it is to transform one distribution into another. This geometric encoding reflects characteristics such as shape, spread, and density. This proves useful in our context, as it allows us to quantify the cost of transforming one hyperedge into another, a measure we argue effectively captures the similarity between group interactions (hyperedges). Figure 1 and Figure 5 in Appendix illustrates how different node features exhibit distinct underlying distribution shapes.

Since our nodes and hyperedges are sets rather than distributions, we use a variant of this embedding called **Sliced Wasserstein Pooling** (Naderializadeh et al., 2021), which is designed not as an embedding of probability distributions themselves, but rather as an embedding of sets sampled from those distributions. In short, Sliced Wasserstein Pooling encodes a set of points by measuring, in an efficient way, how different they are positioned compared to a set of reference points. The complete algorithm as used in our model is described in the following section.

## 4 WASSERSTEIN HYPERGRAPH NEURAL NETWORK

Our Wasserstein Hypergraph Neural Network follows the two-stage framework, by sending information from nodes to hyperedges and vice versa. For simplicity, this section only describes the nodes-to-hyperedges mechanism, as the hyperedge-to-node operation is entirely symmetrical. The pipeline is visually depicted in Figure 2. For readability, the algorithm is presented sequentially for each hyperedge. However, for efficiency, our implementation processes all hyperedges in parallel.

ALGORITHM 1: One Layer of Wasserstein Hypergraph Neural Network[2]

1: **input:** *node features $X \in \mathbb{R}^{N \times L}$ of hypergraph $\mathcal{H}$ and ref. distribution q*
2: **output:** *updated node features $\tilde{X}$*

3: **procedure** WHNN($X, \mathcal{H}, q$)

4:      $X_0 \leftarrow X \in \mathbb{R}^{N \times L}$

5:      *# Sample reference sets*
6:      $Q_v, Q_e \leftarrow sample(q) \in \mathbb{R}^{R \times L}$

7:      *# Extract node and edge neighbourhood*
8:      $\mathcal{N}_v, \mathcal{N}_e \leftarrow$ neighbourhoods($\mathcal{H}$)

9:      *# Node to hyperedge*
10:     $X \leftarrow encoder(X) \in \mathbb{R}^{N \times L}$
11:     $Z \leftarrow Wasserstein(X, \mathcal{N}_v, Q_v) \in \mathbb{R}^{E \times L}$

12:     *# Hyperedge to node*
13:     $Z \leftarrow encoder(Z) \in \mathbb{R}^{E \times L}$
14:     $X \leftarrow Wasserstein(Z, \mathcal{N}_e, Q_e) \in \mathbb{R}^{N \times L}$

15:     *# Residual connection*
16:     $\tilde{X} \leftarrow \alpha X + (1 - \alpha) X_0 \in \mathbb{R}^{N \times L}$

17:     **return** $\tilde{X}$

---

[3] For simplicity in handling shapes, we assume encoders that are independent of the hyperedge.

ALGORITHM 2: Wasserstein aggregator

1: **input:** *features $X \in \mathbb{R}^{N \times L}$; list of aggregated neighbourhoods $\mathcal{N}$; samples from reference distribution $Q \in \mathbb{R}^{R \times L}$*
2: **output:** *aggregated neighbourhoods Z*

3: **procedure** WASSERSTEIN($X, \mathcal{N}, Q$)

4:      *# Project entities into slices*
5:      $X \leftarrow X\Theta \in \mathbb{R}^{N \times L}$
6:      *# Sort the reference points on each slice*
7:      $Q \leftarrow$ sort($Q$) $\in \mathbb{R}^{R \times L}$

8:      **for** $i = 1$ **to** $|\mathcal{N}|$

9:          *# Extract elements in the neighbourhood*
10:         $\tilde{X}^{(i)} \leftarrow \{x_j\}_{j \in \mathcal{N}_i}$

11:         *# If $|\tilde{X}^{(i)}| \neq |Q|$ interpolate $\tilde{X}^{(i)}$*
12:         $\tilde{X}^{(i)} \leftarrow$ interp($\tilde{X}^{(i)}$) $\in \mathbb{R}^{R \times L}$

13:         *# Sort the elements of $\mathcal{N}_i$ on each slice*
14:         $\tilde{X}^{(i)} \leftarrow$ sort($\tilde{X}^{(i)}$) $\in \mathbb{R}^{R \times L}$

15:         *# Compute the dist to approx Wass dist*
16:         $Z_i \leftarrow Q - \tilde{X}^{(i)} \in \mathbb{R}^{R \times L}$

17:     *# Combine the slices based on $W \in \mathbb{R}^{R \times L}$*
18:     $Z \leftarrow Z \odot W \in \mathbb{R}^{E \times R \times L}$
19:     $Z \leftarrow$ mean($Z$, axis = 1) $\in \mathbb{R}^{E \times L}$

20:     **return** $Z$

---

First, we will project the node features into a more expressive representation. Each hyperedge is then associated with a probability distribution, with its constituent nodes treated as samples. These distributions are embedded using a Wasserstein-based aggregator to obtain the final hyperedge representations. Then, the hyperedge representations are fed into the hyperedges-to-nodes stage. Below, we elaborate on each of these stages.

**Node encoder.** The goal of this module is to enhance the representation of node features by projecting them into a more informative space. We are experimenting with two types of encoders: an *edge-independent* one where the node is carrying the same representation in each hyperedge it is contained, and an *edge-dependent* one which takes into account pairwise interactions.

The edge-independent encoder is a simple MLP, which is applied in parallel for each node. This way, a node $i$ is characterised by the same feature vector in each hyperedge $e$ it is part of.

$$\tilde{x}_i^e = \text{MLP}(x_i)$$

On the other hand, for the edge-dependent encoder, each node has a different representation in each hyperedge it is part of. To achieve this, for each hyperedge, we are using a Set Attention Block layer (SAB) as introduced in Lee et al. (2019), which propagates the information between each pair of two nodes contained in that hyperedge. The full version of the block acts as follows:

$$z_i^e = \sigma(x_i + \sum_{j \in e} (x_i W_q)(x_j W_k)^T (x_j W_v))$$
$$\tilde{x}_i^e = \sigma(z_i^e + \text{MLP}(z_i^e)),$$

where $\sigma$ denote layer normalisation and $W_k$, $W_q$ and $W_v \in \mathbb{R}^{d \times d}$ are learnable parameters.

**Hyperedges as probability distributions.** Unlike traditional hypergraph approaches that treat a hyperedge as a set of nodes, we model a hyperedge as a probability distribution, with its constituent nodes being samples drawn from that distribution. This way, the hyperedges are not only characterised

by the combination of their elements, but by the regions of the space where their elements are situated. The nodes became prototypes of the hyperedge behaviour.

Concretely, the shape of the hyperedge distributions can reveal different patterns in our data. For example, a hyperedge connecting nodes from two separate clusters forms a bimodal distribution, showing a mix of groups. A hyperedge where nodes are close together in feature space creates a unimodal distribution, reflecting similar, homophilic behavior. If the node features are very similar, the distribution has low variance, indicating a cohesive hyperedge. On the other hand, a hyperedge with more varied nodes produces a wider, more uniform distribution, showing diverse or loosely connected nodes (see Figure 1). This perspective allows us to interpret the internal structure of hyperedges in a more nuanced way, as the geometry of the distribution naturally reflects the characteristics of the node groups it connects. We consider these elements essential to capture; therefore, we design an aggregator with the appropriate inductive bias to do so.

Let's consider $p_i$ the probability distribution where the elements of the hyperedge $e_i$ are sampled from. In other words, we assume each node $v_j \in e_i$ is sampled as $\tilde{x}_j^i \in \mathbb{R}^d \sim p_i$. The goal is to obtain hyperedge embeddings that preserve the geometric information of this underlying distribution, such as spreading, shape etc. See Figure 5 in the Appendix for a visual representation of this structure.

Note that, by treating nodes as samples from an underlying distribution, we assume that other unobserved nodes drawn from the same distribution are likely to belong to the same hyperedge. This probabilistic interpretation proved to be powerful for set representation learning (Naderializadeh et al., 2021), and our experiments demonstrate that hypergraph models can benefit from it as well.

**Wasserstein aggregator.** Interpreting hypergraphs as a collection of probability distributions allows us to derive more powerful similarity metrics between hyperedges. As shown in the previous section, most of the current hypergraph architectures rely on mean pooling to create hyperedge embeddings from node representations. From a probabilistic perspective, averaging compares distributions only based on their means. For complex data distributions, this approach fails to capture the underlying geometry. While models relying on summation, such as Deep Sets (Zaheer et al., 2017) have been proven to be universal approximators, they heavily rely on the internal node encoder (an MLP) to map the features into a space where first-order statistics like mean effectively approximate the distribution. In the hypergraph setting, where multiple sets interact in complex ways, this is hard to achieve.

This motivates us to adopt Sliced Wasserstein Pooling (Naderializadeh et al., 2021) to encode the hyperedge distributions. Concretely, for each hyperedge $e$, given the node embeddings of all the nodes in the hyperedge $\{\tilde{x}_i^e\}_{i \in e}$, we are aggregating them using the Sliced Wasserstein Pooling to obtain a vectorial hyperedge representation: $h_e = \mathcal{SWP}(\{\tilde{x}_i^e\}_{i \in e})$. The algorithm works as follows:

**Step 1**: Select a reference hyperedge distribution $q$ and sample $N$ points $\{y_i\}_{i=1}^N \sim q$. Choose a set of directions $\{\theta_l\}_{l=1}^L$ with $\theta_l \in \mathbb{R}^{d \times 1}$ used as projection slices in the pooling process. Note that, in order to obtain comparable embeddings across the entire hypergraph, we share the same reference distribution and the same set of slices for all hyperedges.

**Step 2**: Project each node representation $\tilde{x}_i^e$ into each slice $\theta_l$ as follow: $z_i^{e,\theta_l} = (\tilde{x}_i^e)^T \theta_l \in \mathbb{R}$. Since the algorithm requires the same number of sampled nodes from both the hyperedge distribution and the reference, when the cardinality of the hyperedge $|e| \neq N$, we increase/decrease the number of nodes in $e$ using linear interpolation $z_i^{e,\theta_l} \leftarrow \text{interp}(z_i^{e,\theta_l}, N)$ (see Appendix C for details).

**Step 3**: For each hyperedge, for each slice, compute the distance between the node representations and the reference points. $h_e^{\theta_l} = ||z_{\pi(i)}^{e,\theta_l} - y_{\tilde{\pi}(i)}||$, where $z_\pi^{e,\theta_l}$ and $y_{\tilde{\pi}}$ represent the vectors in sorted order. The final hyperedge embedding is obtained as a concatenation of these embeddings: $h_e = ||\{h_e^{\theta_l}\}_{l=1}^L$. Additional post-processing can be applied to project this representation into a vector of the desired dimension.

The process is also described in Algorithm 2. The directions $\theta_l$ and the reference distribution can be either fixed at initialization or trained as learnable parameters.

Intuitively, each hyperedge is represented by a vector which measures how difficult it is to transform the hyperedge distribution into the reference distribution. Note that these reference distributions act only as shared anchors, similar to the origin in Euclidean space (see Appendix A for a detailed

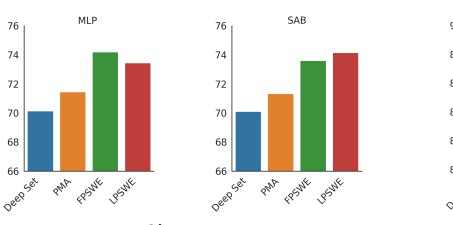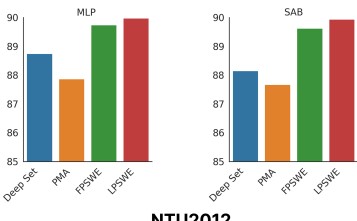

**Citeseer**         **NTU2012**

Figure 3: **Ablation on the importance of Wasserstein aggregator**. We test two versions of Sliced Wasserstein Pooling: with fixed (FPSWE) or learnable (LPSWE) reference distribution. Wasserstein aggregators outperform both Deep Sets and PMA, commonly used inside hypergraph models.

discussion). The true strength of Wasserstein embeddings is not observed in isolation, but lies in their ability to capture the relative distances between different entities (nodes/hyperedges).

Following the theoretical properties of Sliced Wasserstein Pooling (Naderializadeh et al., 2021), *the Euclidean distance between two hyperedge embeddings computed using the Wasserstein aggregator approximate the Wasserstein distance between the hyperedges*: $\|\phi(e_i) - \phi(e_j)\|_2 \approx \mathcal{SW}_2(p_i, p_j)$. In other words, the distance between the embeddings measure the cost of transforming one hyperedge distribution into another. Similarly, the Euclidean distance between two node embeddings measures how easy it is to map one node neighbourhood into the other node neighbourhood.

To understand the practical advantage of using this aggregator, in Figure 1 we computed the Euclidean distance between the Wasserstein embedding of two hyperedges and the Euclidean distance between the mean-based representation of the hyperedges. While the mean aggregators can't distinguish between the three scenarios, the Wasserstein aggregator is able to capture these geometric differences such as the difference in shape and spreading of the nodes.

This information is particularly important in hypergraph learning, as it reflects the extent of change required to transform the characteristics of one group to resemble those of another. In the context of node classification, this means that if the neighbourhoods of two nodes are similar in distribution, the nodes are likely to share the same label. In contrast, average pooling tends to assign same labels to nodes whose neighbourhoods have similar average characteristics, regardless of the form.

**Edge to node step.** For simplicity, we only described in detail the first stage of the framework, which sends messages from nodes to hyperedges. The second stage of the framework, which creates node representation by aggregating the information from neighbouring hyperedges, is done in a similar way, only with different parameters. In conclusion, we not only capture the structural relationship between hyperedges, but also the structural relationship between nodes' neighbourhood.

## 5 EXPERIMENTS

Our main goal is to understand to what extent Wasserstein aggregation is beneficial for hypergraph neural networks. Additionally, we investigate how the choice of node encoder (whether edge-dependent or edge-independent) affects overall performance. Finally, we compare our model against a range of strong baseline methods from the existing literature.

**Datasets.** We test our model on the node-classification task. We select ten real-world datasets that vary in domain and scale. These include Cora, Citeseer, Cora-CA, DBLP-CA (Yadati et al., 2019), ModelNet40 (Wu et al., 2015), NTU2012 (Chen et al., 2003) and 20News (Mitchell, 1997). In the Appendix A, we also offer additional results on Senate, Congress (Fowler, 2006) and House (Chodrow et al.). Among the standard benchmarks for hypergraph models (Yadati et al., 2019), we omitted Pubmed due to the high percentage of isolated nodes (80.5%), which makes the relational processing unnecessary. We follow the training procedures employed by Wang et al. (2022), randomly splitting the data into 50% training, 25% validation and 25% test samples.

**Importance of Wasserstein aggregator.** Our main contribution consists of introducing Sliced Wasserstein Pooling as a powerful aggregator for hypergraph networks. While most of the existing methods use variations of the sum pooling to aggregate the information from each neighbourhood,

Table 2: **Performance on a collection of hypergraph datasets.** Our model using SWP as a node and hyperedge aggregator shows superior results. We test our model in both its variants: with edge-independent (MLP) and edge-dependent encoder (SAB). Both options are exhibiting competitive performance. We mark the **first**, second and *third* best performing models for each dataset.

| Name | Cora | Citeseer | Cora_CA | DBLP_CA | ModelNet40 | NTU2012 | 20News |
|------|------|----------|---------|---------|------------|---------|--------|
| HCHA | $79.14 \pm 1.02$ | $72.42 \pm 1.42$ | $82.55 \pm 0.97$ | $90.92 \pm 0.22$ | $94.48 \pm 0.28$ | $87.48 \pm 1.87$ | $80.33 \pm 0.80$ |
| HNHN | $76.36 \pm 1.92$ | $72.64 \pm 1.57$ | $77.19 \pm 1.49$ | $86.78 \pm 0.29$ | $97.84 \pm 0.25$ | $89.11 \pm 1.44$ | $81.35 \pm 0.61$ |
| HyperGCN | $78.45 \pm 1.26$ | $71.28 \pm 0.82$ | $79.48 \pm 2.08$ | $89.38 \pm 0.25$ | $75.89 \pm 5.26$ | $56.36 \pm 4.86$ | $81.05 \pm 0.59$ |
| HyperGNN | $79.39 \pm 1.36$ | $72.45 \pm 1.16$ | $82.64 \pm 1.65$ | $91.03 \pm 0.20$ | $95.44 \pm 0.33$ | $87.72 \pm 1.35$ | $80.33 \pm 0.42$ |
| AllDeepSets | $76.88 \pm 1.80$ | $70.83 \pm 1.63$ | $81.97 \pm 1.50$ | $91.27 \pm 0.27$ | $96.98 \pm 0.26$ | $88.09 \pm 1.52$ | $81.06 \pm 0.54$ |
| AllSetTransformers | $78.58 \pm 1.47$ | $73.08 \pm 1.20$ | $83.63 \pm 1.47$ | $91.53 \pm 0.23$ | *98.20 ± 0.20* | $88.69 \pm 1.24$ | *81.38 ± 0.58* |
| UniGCNII | $78.81 \pm 1.05$ | $73.05 \pm 2.21$ | $83.60 \pm 1.14$ | $91.69 \pm 0.19$ | $98.07 \pm 0.23$ | $89.30 \pm 1.33$ | $81.12 \pm 0.67$ |
| ED-HNN | 80.31 ± 1.35 | *73.70 ± 1.38* | *83.97 ± 1.55* | 91.90 ± 0.19 | $97.75 \pm 0.17$ | *89.48 ± 1.87* | $81.36 \pm 0.55$ |
| WHNN_MLP | *79.84 ± 1.56* | 74.79 ± 1.19 | 84.12 ± 1.94 | *91.73 ± 0.24* | 98.47 ± 0.19 | **90.87 ± 1.59** | **81.83 ± 0.68** |
| WHNN_(I)SAB | **80.72 ± 1.96** | **74.92 ± 1.60** | **84.62 ± 1.77** | **91.99 ± 0.33** | **98.54 ± 0.21** | 90.68 ± 1.68 | 81.42 ± 0.60 |

our Wasserstein aggregator presents a more in-depth understanding of the neighbourhood distribution, having the inductive bias to capture subtle differences, such as the difference in shape or spread.

To understand to what extent this is contributing to a better hypergraph representation, we design an ablation study in which we keep the underlying architecture fixed and only modify the aggregator. Concretely, we are using as aggregators either Deep Set (as used by AllDeepSet (Chien et al., 2022) and ED-HNN (Wang et al., 2022) models) or the PMA module (as in AllSetTransformer (Chien et al., 2022) model). For our Wasserstein aggregator, we experiment with both a fixed-reference distribution (denoted as FPSWE) or with learnable reference distribution (LPSWE). For a robust evaluation, we are comparing these aggregators using both the edge-independent (MLP) and the edge-dependent encoder (SAB). The results on Citeseer and NTU2012 datasets are reported in Figure 3.

Regardless of the encoder and the dataset we are testing on, both Wasserstein aggregators are consistently outperforming both the Deep Sets and the PMA aggregators by a significant margin. A learnable reference seems to be beneficial however, the improvement is generally marginal. Additional experiments on other datasets show a similar trend and are provided in the Appendix A.

**Importance of edge-dependent encoder.** The node and hyperedge encoder transforms features into a space where their distribution within each hyperedge captures meaningful information about the group. As stated in the model description, we equipped our model with two types of encoders. An edge-independent module represented by an MLP, and an edge-dependent encoder represented by a self-attention block (SAB). While the MLP is processing information independently for each node/hyperedge, SAB is capturing pairwise interactions between nodes/hyperedges sharing a neighbourhood. The results in Figure 3 and Table 2 show that encoder performance varies, with the edge-dependent encoder sometimes outperforming others, particularly on more homophilic datasets. However, this increased power comes at the cost of higher computational and memory requirements, as the edge-dependent encoder must store all incident (node, hyperedge) pairs. To mitigate this on larger datasets such as 20News and DBLP, we replace the SAB block with the ISAB low-rank approximation introduced by Lee et al. (2019).

**Comparison with baselines.** In Table 2, we are comparing against a series of hypergraph networks from the literature. With respect to aggregation strategies, HNHN (Dong et al., 2020), Hyper-GNN (Feng et al., 2019), AllDeepSets (Chien et al., 2022), UniGCNII (Huang & Yang, 2021a) and ED-HNN (Wang et al., 2022) use variations of Deep Sets to aggregate the information, HyperGCN (Yadati et al., 2019) uses a max aggregator, while HCHA (Bai et al., 2021) and AllSetTransformer (Chien et al., 2022) use an attention-based weighted summation. Regardless of the encoder used, our model consistently obtains top results, outperforming the other methods on all datasets. This demonstrates the advantages of using Wasserstein aggregators for higher-order processing. While we integrated this aggregator into a standard two-stage framework, many existing models from the literature can be adopted to take advantage of this type of geometric-inspired aggregation.

**Implementation details.** We train our models using Adam for 500 epochs, on a single GPU NVIDIA Quadro RTX 8000 with 48GB of memory. Each model is trained 10 times with different random splits and initialisations. For the non-ablation experiments, the results are based on the results of Wang et al. (2022), while the performance of our model represent the best performing model obtained

during hyper-parameter optimisation (see Appendix C). For the ablation study, the architecture is fixed to ensure a fair comparison. We use a number of Wasserstein slices equal to the hidden dimension, and we experiment with both learning the reference set or not. The reference is set to a uniform distribution and we vary the number of points sampled. In Appendix A, we offer additional experiments demonstrating that, as expected, the type of reference distribution is not essential, while the number of sampled points should be large enough to cover the complexity of the hyperedge set. The computational complexity is reported in Appendix D.

These experimental results show that aggregating node and hyperedge neighbourhoods using Sliced Wasserstein Pooling is highly effective for hypergraph processing, the Wasserstein aggregator consistently outperforming standard methods like Deep Sets and PMA.

## 6 CONCLUSION

In this work, we introduce Wasserstein Hypergraph Neural Networks (WHNN), a model for processing hypergraph structures. The model relies on Sliced Wasserstein Pooling to aggregate the nodes into hyperedge representations and vice versa. This design choice, inspired by optimal transport literature, enables us to capture more information about the internal structure of the neighbourhoods, preserving more geometric relations between elements. The experimental results on various datasets demonstrate that this Wasserstein aggregator is effective for modelling higher-order interactions, outperforming traditional aggregators, making WHNN a promising tool for hypergraph representation learning.

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

# Appendix: Wasserstein Hypergraph Neural Network

This appendix contains details related to our model, including additional comparisons and ablations, potential limitations and future work, details on the hyperparameters used in our experiments and derivation of the computational complexity. In addition to this, we also include the code to ensure reproducibility. The content is structured as follows:

- **Section A** presents results on three more datasets and additional experiments used as ablation for our model.
- **Section B** highlights a series of potential limitations that can be addressed to improve the current work, together with a discussion on potential future work.
- **Section C** presents the list of hyperparameters used in our experiments.
- **Section D** derives the computational complexity of our model.

## A ADDITIONAL EXPERIMENTS

**Experiments on additional datasets.** Due to space constrain, in the main paper, we show results on seven benchmarks usually used in the hypergraph literature. Here, we provide additional results on three more datasets, Senate, Congress (Fowler, 2006) and House (Chodrow et al.). Compared to the previous ones, for these datasets, the nodes are not equipped with features, so we adopt the usual setup in which synthetic features are generated using Gaussian noise (Wang et al., 2022). However, this limitation of the benchmarks makes it harder to interpret or understand the input space.

The results in Table 3 show a consistent trend with the other benchmarks: even when the feature space is synthetically generated, the Wasserstein aggregator enhances the representations and yields improved performance.

Table 3: **Performance comparison on Congress, Senate, and House datasets.** The WHNN model, which improves the message passing hypergraph architecture with a Wasserstein aggregator, leads to better results, clearly overcoming the DeepSet-based models.

| Model | Congress | Senate | House |
|---|---|---|---|
| HCHA | $90.43 \pm 1.20$ | $48.62 \pm 4.41$ | $61.36 \pm 2.53$ |
| HNHN | $53.35 \pm 1.45$ | $50.93 \pm 6.33$ | $67.80 \pm 2.59$ |
| HyperGCN | $55.12 \pm 1.96$ | $42.45 \pm 3.67$ | $48.32 \pm 2.93$ |
| HyperGNN | $91.26 \pm 1.15$ | $48.59 \pm 4.52$ | $61.39 \pm 2.96$ |
| AllDeepSets | $91.80 \pm 1.53$ | $48.17 \pm 5.67$ | $67.82 \pm 2.40$ |
| AllSetTransformer | $92.16 \pm 1.05$ | $51.83 \pm 5.22$ | $69.33 \pm 2.20$ |
| UniGCNII | $94.81 \pm 0.81$ | $49.30 \pm 4.25$ | $67.25 \pm 2.57$ |
| ED-HNN | $95.00 \pm 0.99$ | $64.79 \pm 5.14$ | $72.45 \pm 2.28$ |
| WHNN_MLP | $\mathbf{95.67 \pm 0.90}$ | $66.48 \pm 3.56$ | $\mathbf{72.66 \pm 1.26}$ |
| WHNN_(I)SAB | $95.42 \pm 0.99$ | $\mathbf{67.04 \pm 4.80}$ | $72.04 \pm 1.78$ |

**Importance of Wasserstein aggregator.** In the main paper, we included ablation studies on Citeseer and NTU datasets. Here we report additional results for Cora_CA and ModelNet40 datasets (Figure 4) together with the numerical results for all experiments (Table 4).

For each experiment, we kept the architecture fixed and modified the aggregator used in the two stages to be either Deep Set, PMA, or the learnable (LPSWE) or fixed (FPSWE) Wasserstein aggregator. The results are similar across the datasets, with Wasserstein Pooling proving to be beneficial compared to Deep Sets and PMA. In terms of encoder type, we noticed that, in some cases, for a fixed architecture, SAB tends to model the distribution better than MLPs.

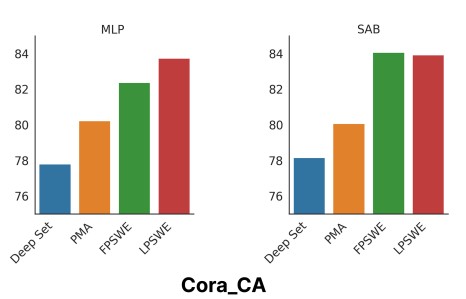 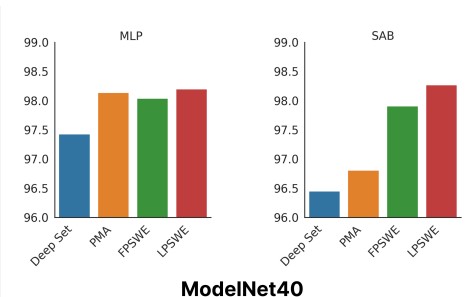

Figure 4: **Additional results for the ablation study on the importance of Wasserstein aggregator for hypergraph representation learning** Cora_CA and ModelNet datasets. FPSWE denotes the Wasserstein aggregator with fixed reference while LPSWE denotes the Wasserstein aggregator with learnable reference distribution. Regardless of the encoder used to project the nodes and hyperedges, the Wasserstein aggregators outperform both the Deep Sets and PMA commonly used inside hypergraph models.

Table 4: **Numerical results for the ablation study** comparing DeepSet, PMA and our Wasserstein aggregator with fixed (FPSWE) or learnable (LPSWE) references.

| Model | MLP encoder | | | | SAB encoder | | | |
|---|---|---|---|---|---|---|---|---|
| | Citeseer | NTU | Cora_ca | ModelNet | Citeseer | NTU | Cora_ca | ModelNet |
| DeepSet | $70.14 \pm 0.69$ | $88.75 \pm 1.88$ | $77.81 \pm 2.03$ | $97.43 \pm 0.29$ | $70.10 \pm 0.57$ | $88.15 \pm 1.32$ | $78.17 \pm 1.02$ | $96.45 \pm 0.42$ |
| PMA | $71.45 \pm 0.48$ | $87.87 \pm 1.79$ | $80.23 \pm 0.81$ | $98.14 \pm 0.26$ | $71.33 \pm 0.86$ | $87.67 \pm 1.53$ | $80.08 \pm 1.35$ | $96.81 \pm 0.33$ |
| FPSWE | $74.20 \pm 0.66$ | $89.74 \pm 1.65$ | $82.39 \pm 1.27$ | $98.04 \pm 0.32$ | $73.60 \pm 0.99$ | $89.62 \pm 1.61$ | $84.07 \pm 1.23$ | $97.91 \pm 0.24$ |
| LPSWE | $73.45 \pm 0.81$ | $89.98 \pm 1.62$ | $83.75 \pm 1.74$ | $98.20 \pm 0.27$ | $74.15 \pm 0.99$ | $89.94 \pm 1.59$ | $83.93 \pm 1.68$ | $98.27 \pm 0.31$ |

**Influence of the reference distribution and the number of samples.** For our nodes and hyperedge embeddings, the reference distribution only acts as a common anchor, similar to the origin in an Euclidean space. On its own, each set embedding (node/hyperedge representation) contains information about how different the underlying distribution is compared to the same reference distribution. However, when computing the relative distance between two sets (two nodes, two hyperedges), the reference distribution cancels out, and we obtain information about how one hyperedge can be transformed into another (regardless of the shape and characteristics of the reference distribution). Because of that, we expect the choice of reference distribution to have minimal impact on the performance, mostly attributed to numerical stability.

To validate this, we design a series of experiments in which we modify the shape of the reference distribution and the number of samples used to represent it.

In the first experiment, we pick the reference distribution to be either uniform, Gaussian, Poisson or a learned distribution. For the learnable distribution, we consider the samples from the distribution to be learnable parameters.

The results in Table 5 suggest that the choice of distribution has little effect, with only slight variations observed in the NTU datasets and for the learnable distribution, differences likely due to computational stability issues.

Table 5: **Ablation study comparing performance when varying the type of reference distributions and the number of points sampled from them.** Since the reference distribution serves only as a shared anchor for all sets, it has little impact on final accuracy. The empirical results on three datasets confirm this intuition.

(a) Ablation study on Citeseer dataset.

| Distr. | 2 ref | 5 ref | 10 ref | 25 ref | 50 ref |
|---|---|---|---|---|---|
| Uniform | $74.55 \pm 1.68$ | $74.88 \pm 1.59$ | $74.72 \pm 1.63$ | $74.69 \pm 1.77$ | $74.84 \pm 1.73$ |
| Normal | $74.51 \pm 1.73$ | $74.83 \pm 1.61$ | $74.71 \pm 1.65$ | $74.71 \pm 1.79$ | $74.81 \pm 1.72$ |
| Poisson | $74.63 \pm 1.54$ | $74.81 \pm 1.53$ | $74.70 \pm 1.66$ | $74.66 \pm 1.79$ | $74.83 \pm 1.73$ |
| Learnable | $74.62 \pm 1.55$ | $74.60 \pm 1.41$ | $74.85 \pm 1.54$ | $74.68 \pm 1.75$ | $74.85 \pm 1.69$ |

(b) Ablation study on Cora_CA dataset

| Distr. | 2 ref | 5 ref | 10 ref | 25 ref | 50 ref |
|---|---|---|---|---|---|
| Uniform | $84.38 \pm 1.51$ | $84.74 \pm 1.60$ | $84.63 \pm 1.63$ | $84.56 \pm 1.71$ | $85.03 \pm 1.82$ |
| Normal | $84.19 \pm 1.79$ | $84.69 \pm 1.57$ | $84.53 \pm 1.65$ | $84.50 \pm 1.80$ | $84.75 \pm 1.79$ |
| Poisson | $84.12 \pm 1.80$ | $84.51 \pm 1.61$ | $84.63 \pm 1.66$ | $84.49 \pm 1.73$ | $84.70 \pm 1.71$ |
| Learnable | $84.29 \pm 1.94$ | $84.51 \pm 1.49$ | $84.50 \pm 1.35$ | $84.62 \pm 1.63$ | $84.62 \pm 1.75$ |

(c) Ablation study on NTU dataset.

| Distr. | 2 ref | 5 ref | 10 ref | 25 ref | 50 ref |
|---|---|---|---|---|---|
| Uniform | $90.19 \pm 1.38$ | $90.21 \pm 1.60$ | $90.67 \pm 1.17$ | $90.47 \pm 1.31$ | $90.80 \pm 1.21$ |
| Normal | $90.18 \pm 1.37$ | $90.07 \pm 1.57$ | $90.59 \pm 1.50$ | $91.01 \pm 1.46$ | $90.41 \pm 1.47$ |
| Poisson | $90.19 \pm 1.38$ | $90.25 \pm 1.61$ | $90.50 \pm 1.49$ | $90.39 \pm 1.02$ | $90.41 \pm 1.87$ |
| Learnable | $90.51 \pm 1.39$ | $90.50 \pm 1.38$ | $90.52 \pm 1.26$ | $90.62 \pm 1.37$ | $90.58 \pm 1.06$ |

In the second set of experiments, we run the same setup as before, but modify the number of sampled points from the reference distribution. If all sets have the same cardinality, the standard approach is to select a number of reference points equal to this cardinality (thus avoiding the need for linear interpolation). However, this is not possible in the hypergraph domain, where hyperedges tend to have various cardinalities. Thus, we expect the model to perform well as long as we pick the number of reference points to be comparable to most of the cardinalities. The experiments in Table 5 show a small drop in performance for very few reference points (when $M = 2$ for Citeeser and Cora_CA datasets, and $M \in \{2, 5\}$ for NTU), with comparable performance otherwise.

**Additional details for the synthetic example.** In Figure 1 of the main paper, the three inputs were synthetically generated by sampling points from distributions with different configurations. The reported distances were computed empirically using our implementation. Specifically, for the Wasserstein distance, the reported value corresponds to the $L_2$ distance between the Wasserstein embeddings produced by our aggregator, which serves as an approximation of the true Wasserstein distance.

## B  LIMITATIONS AND FUTURE WORK

Consistent with findings in set representation learning, employing a more sophisticated aggregator, such as the one introduced in this work, inevitably increases computational overhead. Although this is not an issue for current hypergraph benchmarks, it may become challenging for larger hypergraphs, which could necessitate using fewer reference points or incorporating sampling strategies to control computational cost. While further optimisation of Wasserstein embedding computation is still needed, any such advances would be highly beneficial for our model.

As discussed in the main paper, we treat the neighbourhood of each node as a sample from an underlying probability distribution. This approach assumes that any additional nodes drawn from

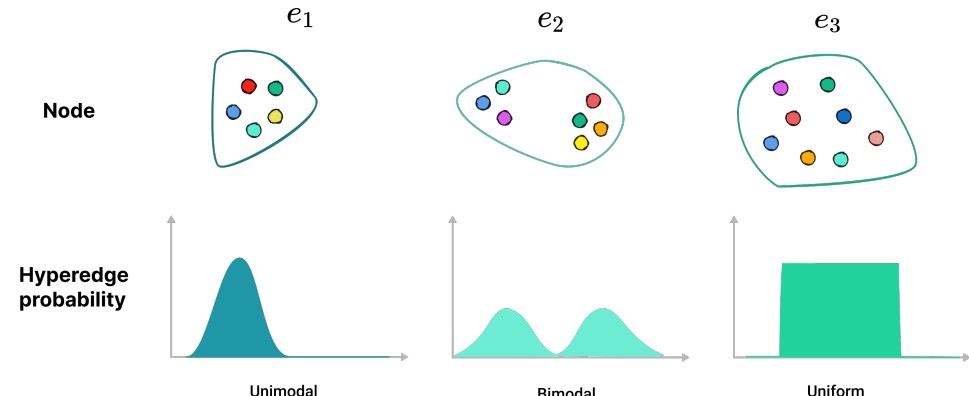

Figure 5: Diagram showing how different geometric arrangements of node embeddings correspond to distributions with varying shapes and spreads.

this distribution should belong to the same neighbourhood as the observed ones. This aligns with the intuition that elements within a group should share common characteristics. While the datasets we used support this assumption, there may be real-world scenarios where it does not hold. Our model relies solely on the node encoder to project features into a space where the assumption is approximately valid.

Moreover, due to this continuous view of the neighbourhood (as a distribution of probability) together with the interpolation step, the current model may lose information about the exact cardinality of the neighbourhoods. In situations where neighbourhood size is important, we recommend encoding it as an explicit feature. However, we mention that this is an issue we share with the mean-based pooling algorithms.

The main goal of this paper is to highlight the benefits of using geometrically-inspired poolings for aggregating neighbourhood information in hypergraphs. While we focused entirely on hypergraphs, a similar idea can be applied on graph neural networks or other topological structures to aggregate messages coming from each node's neighbourhood. As future work, it would be interesting to see to what extent these models can benefit from Wasserstein aggregators.

Moreover, while the proposed model integrates the Wasserstein aggregator into a standard two-stage pipeline, several other architectures, such as ED-HNN which uses summation as an aggregator, might benefit from adopting it. We are leaving this investigation as future work.

## C  IMPLEMENTATION DETAILS

The results reported in Table 2 of the main paper are obtained using random hyperparameter tuning. We report here the range of parameters that we searched for. Table 6 and Table 7 contain the best hyperparameter configuration for the WHNN_MLP model and WHNN_SAB. We depict in bold the parameters specific to the Wasserstein aggregator, in italic the parameters specific to the SAB encoder, while the rest of them are the standard parameters used in the two-stage hypergraph models. In our experiment, we search for the following hyperparameters:

- num_ref: number of elements sampled from the reference distribution $\{5, 10, 25, 50\}$
- learnable_W: choose between learning or not the reference distribution $\{\text{True}, \text{False}\}$
- heads: number of heads used by the SAB block $\{1, 2, 4\}$
- MLP_layers: number of layers in all MLPs used $\{0, 1, 2\}$
- MLP_hid: number of hidden units in all MLPs used. This is also the number of slices used by Wasserstein aggregator. $\{128, 256, 512\}$
- MLP2_layers: using or not an additional linear projection after the residual connection of each stage $\{0, 1\}$
- Cls_layers: number of layers in the final classifier MLP $\{1, 2\}$

- Cls_hid: number of hidden units in the final classifier MLP $\{96, 128, 256\}$

- self_loops: using or not self loops $\{\text{True}, \text{False}\}$

- dropout: dropout used inside the model $\{0.5, 0.6, 0.7\}$

- in_dropout: dropout used in the begining of the model $\{0.2, 0.5, 0.6, 0.7\}$

- fixed hyperparameters: All models use 1 layer of WHNN, LayerNorm normalisation, the residual coefficient $\alpha$ fixed to $0.5$, and they are trained for 500 epochs with a learning rate of $0.001$.

The reason we included the use of self-loops as an additional hyperparameter stems from the specific probabilistic formulation of our model. In architectures that rely on sum-based aggregators, a self-loop functions essentially as a residual connection; however, this equivalence does not hold in our case. In our construction, the self-loop corresponds to a probability distribution concentrated at a single point, which can be undesirable in certain scenarios. For this reason, we decided to evaluate both configurations.

**Details on the linear interpolation.** The general purpose of the interpolation step in Algorithm 2 is to *drop* or *invent* points when the cardinality is not matching the desired one. In our experiments, we apply simple linear interpolation to samples from 1-dimensional distributions (the slices). More precisely, for a hyperedge containing $K$ sample points, we treat these points as evaluations of the inverse CDF at $K$ uniformly spaced quantiles in $[0, 1]$. To obtain a representation with $R$ points instead, we evaluate the inverse CDF at $R$ new uniformly spaced quantiles by linearly interpolating between the original $K$ values. This produces an $R$-point discretization that is consistent with the distribution implied by the original samples. During the development we also experimented with other interpolation options (as supported by python [3]) including the nearest neighbour one which just copy points or drop them depending on the need, but we haven't noticed substantial differences in performance.

Table 6: The best configuration of hyperparameters used by our model WHNN_MLP on all tested datasets. We mark with bold the parameters that are specific to the Wasserstein aggregator.

| Parameter | Cora | Citeseer | Cora_CA | DBLP_CA | ModelNet40 | NTU2012 | 20News |
|---|---|---|---|---|---|---|---|
| **num_ref** | 25 | 10 | 25 | 5 | 50 | 25 | 25 |
| **learnable_W** | True | False | True | True | False | False | False |
| MLP_layers | 1 | 2 | 2 | 2 | 1 | 1 | 0 |
| MLP2_layers | 0 | 0 | 1 | 0 | 0 | 1 | 0 |
| MLP_hid | 128 | 256 | 256 | 512 | 256 | 512 | 512 |
| Cls_layers | 1 | 1 | 1 | 2 | 2 | 2 | 2 |
| Cls_hid | 256 | 128 | 96 | 96 | 96 | 96 | 96 |
| self_loops | True | True | True | True | True | False | False |
| dropout | 0.7 | 0.5 | 0.6 | 0.7 | 0.5 | 0.5 | 0.5 |
| in_dropout | 0.7 | 0.5 | 0.6 | 0.7 | 0.2 | 0.2 | 0.2 |

## D COMPUTATIONAL COMPLEXITY

In this section, we first derive the theoretical complexity for both versions of our Wasserstein Hypergraph Neural Network: using the edge-independent encoder (WHNN_MPN) and using the edge-dependent encoder (WHNN_SAB).

For the theoretical analysis, we present the complexity for a hypergraph with $N$ nodes, $M$ hyperedges, the maximum cardinality of a hyperedge $K_e$, the maximum number of hyperedges a node is part of $K_v$ and $R$ number of reference points sampled from the reference distribution.

Regarding the encoders, the edge-independent one (MLP) has a complexity of $O(N)$ while the edge-dependent one (SAB) has complexity $O(M \times K^2)$ due to the pairwise exchange of messages ($K^2$) inside each hyperedge ($M$).

---

[3]https://docs.scipy.org/doc/scipy/reference/generated/scipy.interpolate.interp1d.html

Table 7: The best configuration of hyperparameters used by our model WHNN_SAB on all tested datasets. We mark with bold the parameters that are specific to the Wasserstein aggregator and with italic the parameters that are specific to the SAB encoder.

| Parameter | Cora | Citeseer | Cora_CA | DBLP_CA | ModelNet40 | NTU2012 | 20News |
|---|---|---|---|---|---|---|---|
| **num_ref** | 10 | 5 | 50 | 5 | 25 | 25 | 5 |
| **learnable_W** | True | False | False | False | False | False | True |
| *heads* | 2 | 4 | 1 | 4 | 1 | 2 | 2 |
| MLP_layers | 2 | 2 | 2 | 1 | 1 | 2 | 2 |
| MLP2_layers | 0 | 0 | 1 | 1 | 0 | 0 | 0 |
| MLP_hid | 128 | 256 | 128 | 256 | 256 | 512 | 512 |
| Cls_layers | 1 | 1 | 1 | 2 | 2 | 2 | 2 |
| Cls_hid | 128 | 256 | 128 | 96 | 96 | 96 | 96 |
| self_loops | True | False | True | True | True | True | False |
| dropout | 0.7 | 0.7 | 0.5 | 0.7 | 0.5 | 0.5 | 0.5 |
| in_dropout | 0.7 | 0.7 | 0.5 | 0.7 | 0.2 | 0.2 | 0.2 |

For the Wasserstein aggregator, the complexity for the nodes to hyperedges stage consists of the complexity of the linear interpolation applied for each hyperedge to obtain $R$ points from the set of $K_e$ points representing the hyperedge. After that, all we need to do is an elementwise difference between the interpolated points and the reference points. To sort each hyperedge, the complexity is $O(K_e log K_e)$, and the complexity for interpolating on each sorted hyperedge is $(R \times log K_e)$ for a total of $M \times (R \times log K_e + K_e log K_e)$. Similarly, for the hyperedge to node stage, the complexity is $N \times (R \times log K_v + K_v log K_v)$.

For comparison, the complexity of a Deep Set pooling is $O(M \times K_e + N \times K_v)$

To provide a concrete comparison, we report runtime and memory usage for our best model alongside the DeepSet and PMA baselines on Citeseer (a medium-sized hypergraph) and DBLP_CA (a large hypergraph). While the Wasserstein aggregator introduces additional computational overhead compared to the simplest aggregation methods, the complexity of the aggregator (both V→E and E→V stages) remains comparable to that of the encoder. Inference time (in seconds) for the encoder (Enc) and aggregator (Agg) at each stage (V→E and E→V) measured on a Quadro RTX 8000 GPU.

Table 8: Inference time (in seconds) for encoder (Enc) and aggregator (Agg) at each stage (V→E, E→V) for Citeseer and DBLP datasets.

| Dataset | Stage | WHNN | PMA | DeepSet |
|---|---|---|---|---|
| Citeseer | V-E Enc. | 0.0022 | 0.0025 | 0.0025 |
| | V-E Agg. | 0.0046 | 0.0023 | 0.0010 |
| | E-V Enc. | 0.0025 | 0.0030 | 0.0027 |
| | E-V Agg. | 0.0035 | 0.0025 | 0.0012 |
| DBLP_CA | V-E Enc. | 0.094 | 0.090 | 0.091 |
| | V-E Agg. | 0.083 | 0.0089 | 0.0019 |
| | E-V Enc. | 0.079 | 0.071 | 0.079 |
| | E-V Agg. | 0.059 | 0.0092 | 0.0019 |

