# OpenReview forum: "Wasserstein Hypergraph Neural Network"
_ICLR.cc/2026/Conference — Submitted to ICLR 2026_

### Official Review · Reviewer_qh5o · 2025-10-25

**Soundness:** 3
**Presentation:** 2
**Contribution:** 1
**Rating:** 2
**Confidence:** 4

**Summary:**

The paper introduces the Wasserstein Hypergraph Neural Network, an architecture designed to overcome a key limitation in existing hypergraph neural networks. WHNN proposes a conceptual shift by viewing these neighborhoods as empirical samples from underlying probability distributions. To capture their geometric properties, the model employs Sliced Wasserstein Pooling as its core aggregation mechanism. This design creates embeddings where the Euclidean distance approximates the Sliced Wasserstein distance between the original distributions, thus preserving structural information like shape, spread, and density. Implemented within a standard two-stage message-passing framework, the authors demonstrate that WHNN outperforms traditional aggregation methods and achieves state-of-the-art performance on multiple node classification benchmark datasets.

**Strengths:**

Principled Aggregator Choice: By adopting the distributional perspective, the choice of Sliced Wasserstein Pooling as an aggregator is well-justified. It endows the model with a powerful inductive bias to capture the shape, spread, and density of feature distributions within hyperedges, a capability that mean-based aggregators lack.

Thorough Empirical Validation: The authors conduct a rigorous evaluation across ten datasets, comparing against eight strong baselines. The inclusion of comprehensive ablation studies effectively isolates the contribution of the Wasserstein aggregator, providing convincing evidence for the paper's central claims

**Weaknesses:**

1.	Theoretical Grounding not thoroughly discussed: the paper critiques sum-based aggregators like Deep Sets. Deep Sets is known to be a universal approximator for permutation-invariant functions defined on sets. The authors provide no such theoretical analysis for WHNN. They rely on the properties of SWP established in prior work, stating that their embedding approximates the SW distance, but they do not analyze what this approximation implies for the expressive power of the full HGNN model. This leaves several important theoretical questions unanswered. Does WHNN retain universal approximation capabilities? How does the error introduced by the SWP approximation propagate through multiple layers of message passing? Is the model guaranteed to distinguish between any two non-isomorphic hypergraph neighborhoods - is the aggregation function injective? The literature on SWP embeddings has explored properties like injectivity and bi-Lipschitz continuity, but the authors do not connect these foundational concepts to their hypergraph model. Without this theoretical grounding, the paper's contribution feels more like an engineering result rather than a fundamental advance in understanding hypergraph representation learning.

2.	Potentially Overstated Claims and Interpretation of Results: The paper claims to achieve "top performance" and "top results" across all datasets. While technically true based on the mean accuracy reported in Table 2, this language can be misleading when the margin of victory is small and falls within the standard deviation of a competing method. For example, on the NTU2012 dataset, WHNN-MLP scores 90.87 $\pm$ 1.59, which is statistically very close to ED-HNN's 89.48 $\pm$ 1.87. A more measured tone would be appropriate. More concerning is the interpretation of the ablation studies. The authors initially motivate SWP as a way to reduce the learning burden on the encoder. However, their own results in Figure 3 show that SWP's performance also significantly improves when paired with the more complex SAB encoder. This suggests the initial premise may be a false dichotomy. A more nuanced conclusion, better supported by the data, is that powerful encoders and expressive aggregators are complementary, and their combination is what yields state-of-the-art results.

3.	Omission of Computational Cost and Practicality Analysis: The paper's central claim revolves around achieving superior classification accuracy, but it entirely neglects to discuss the associated computational costs. The literature on SWP confirms its computational overhead, with a complexity of $O(LM \log M)$ for a set of size $M$ and $L$ slices, and often requires a large number of slices ($L$) to be effective. The paper's own complexity analysis in Appendix D confirms this super-linear dependency on neighborhood size.

**Questions:**

1. Check the first point of Weaknesses, some theoretical proofs should be shown in the paper.
2. The Wasserstein aggregator appears significantly more computationally intensive than the simple summation used in baselines like ED-HNN. Could the authors please provide empirical performance metrics for WHNN? This would allow for a fair assessment of the accuracy-efficiency trade-off.
3. Given that the performance gains are sometimes modest (e.g., $\approx 0.4\%$ absolute improvement over ED-HNN on Cora), how do the authors justify the additional complexity of SWP for practical applications where inference speed or training cost are critical constraints?
4. The ablation studies show that both the baseline aggregators (Deep Set, PMA) and the proposed SWP aggregator benefit from a more powerful encoder. This seems to suggest that the initial motivation—that simpler aggregators uniquely "move the complexity... to the initial encoding"-may be an oversimplification. Could the authors please clarify or provide a more nuanced interpretation of these results? It appears a more general conclusion is that better encoders are complementary to better aggregators.
5. Wasserstein distance is extensively used in AI and machine learning. Why this WHNN adds new values?

---

> ### Author Response · Authors · 2025-11-18
>
> Thank you for your thorough review and constructive feedback. We would like to address your questions.
>
> > Check the first point of Weaknesses, some theoretical proofs should be shown in the paper.
>
> We appreciate the reviewer’s thoughtful comments regarding theoretical guarantees such as injectivity, bi-Lipschitzness, and universal approximation. We agree that our current work does not establish such results for WHNN. Our goal in this paper, however, is to introduce Wasserstein‐based techniques into hypergraph representation learning and to demonstrate that they offer a promising alternative to sum-based aggregation. To our knowledge, WHNN is the first attempt to integrate Wasserstein geometry into hypergraph neural networks, and we view this contribution as an initial step toward a theoretically grounded framework.
> Connecting WHNN to the recent theoretical literature on Wasserstein embedding (such as Amir et al. 2025) is a promising direction. These developments offer stronger guarantees and improved computational properties, and they can be incorporated into the WHNN architecture to provide the theoretical foundations the reviewer highlights (e.g., injective embeddings, stability, or universal approximation properties). We are actively interested in extending WHNN along these lines, and we believe our work opens up exactly this research avenue by establishing the feasibility of Wasserstein-based hypergraph message passing. We have included a discussion of these potential extensions in the related work section to highlight how our approach can be further developed.
>
> > The Wasserstein aggregator appears significantly more computationally intensive than the simple summation used in baselines like ED-HNN. Could the authors please provide empirical performance metrics for WHNN? This would allow for a fair assessment of the accuracy-efficiency trade-off.
>
> The reviewer correctly points out that the Wasserstein aggregator is more computationally intensive than a simple summation. As suggested by the theoretical complexity, this difference becomes more pronounced for large datasets, where a higher number of reference points is required to capture the complexity of the node distributions.
>
> To provide a concrete comparison, we report runtime and memory usage for our best model alongside the DeepSet and PMA baselines on Citeseer (a medium-sized hypergraph) and DBLP_CA (a large hypergraph). While the Wasserstein aggregator introduces additional computational overhead compared to the simplest aggregation methods, the complexity of the aggregator (both V→E and E→V stages) remains comparable to that of the encoder.
>
> Inference time (in seconds) for the encoder (Enc) and aggregator (Agg) at each stage (V→E and E→V) measured on a Quadro RTX 8000 GPU:
>
> | Method  | Citeseer       |             |             |             | DBLP       |             |             |             |
> |---------|----------------|-------------|-------------|-------------|------------|-------------|-------------|-------------|
> |         | V-E Enc.       | V-E Agg.    | E-V Enc.    | E-V Agg.    | V-E Enc.   | V-E Agg.    | E-V Enc.    | E-V Agg.    |
> | WHNN    | 0.0022         | 0.0046      | 0.0025      | 0.0035      | 0.094      | 0.083       | 0.079       | 0.059       |
> | PMA     | 0.0025         | 0.0023      | 0.0030      | 0.0025      | 0.090      | 0.0089      | 0.071       | 0.0092      |
> | DeepSet | 0.0025         | 0.0010      | 0.0027      | 0.0012      | 0.091      | 0.0019      | 0.079       | 0.0019      |
>
> This table is also included in the appendix and is referenced in the experimental section of the main paper.
>
> > Given that the performance gains are sometimes modest (e.g.,  absolute improvement over ED-HNN on Cora), how do the authors justify the additional complexity of SWP for practical applications where inference speed or training cost are critical constraints?
>
> Indeed, there is a tradeoff between the complexity of the aggregator and training or inference cost. In current hypergraph benchmarks, this is generally not a limiting factor due to the relatively small hypergraph sizes. If computational constraints become critical, one can reduce the number of reference points in Sliced Wasserstein Pooling (SWP) to achieve a favourable speed–accuracy tradeoff. As demonstrated in the ablation studies (Table 5), varying the number of reference points has only a minor impact in our setting, and thus does not pose a major complexity concern. Nonetheless, hyperedge cardinality still exerts an independent influence.
>
> We also acknowledge that for simpler node configurations in feature space, sum-based aggregators may be sufficient. However, both theoretical and empirical results from the set-representation learning literature highlight the limitations of simple aggregators, supporting the potential benefits of SWP in capturing richer structural information within hyperedges.

---

> > ### Author Response · Authors · 2025-11-18
> >
> > > The ablation studies show that both the baseline aggregators (Deep Set, PMA) and the proposed SWP aggregator benefit from a more powerful encoder. This seems to suggest that the initial motivation—that simpler aggregators uniquely "move the complexity... to the initial encoding"-may be an oversimplification. Could the authors please clarify or provide a more nuanced interpretation of these results? It appears a more general conclusion is that better encoders are complementary to better aggregators.
> >
> > The reviewer's comment is indeed in line with our discussion in the experimental part (Importance of edge-dependent encoder). As mentioned there,  we agree that the results suggest that powerful encoders and expressive aggregators, such as SWP, are complementary rather than alternatives. This complementarity is particularly evident in homophilic datasets, where the combination of a sophisticated encoder (SAB) and SWP yields the strongest performance, while in the more heterophilic setup (such as NTU, 20News, Congress, House) the simplicity of MLPs is sometimes preferred.
> >
> > We apologize for the oversimplification in the background section, which was based solely on observations from related work (Kothapalli et al) and does not align with our empirical results. We have revised this statement in the manuscript to avoid any potential confusion.
> >
> > > Wasserstein distance is extensively used in AI and machine learning. Why this WHNN adds new values?
> >
> > We acknowledge that Wasserstein-based embeddings have been explored in prior work over the past four years. However, as the reviewer also noted, our paper is the first to integrate Wasserstein pooling into hypergraph representation learning. We believe that this type of aggregator offers a promising new direction for improving message aggregation within hypergraph architectures. Historically, classical ideas such as Deep Sets, Set Transformers, convolutional and attentional mechanisms, and even message passing were initially developed for sets or graphs and only later adapted to hypergraphs, where they became influential components of the literature. In a similar spirit, our work takes an established concept and extends it into a domain where it has not yet been explored, thereby contributing to the field.

---

> > ### Comment · Reviewer_qh5o · 2025-11-28
> >
> > I thank the authors for the response.  Wasserstein distance is used extensively in AI and machine learning.  Without theoretical guarantees, the results are ad hoc in some sense.

---

### Official Review · Reviewer_N3sh · 2025-10-26

**Soundness:** 4
**Presentation:** 4
**Contribution:** 4
**Rating:** 8
**Confidence:** 4

**Summary:**

This paper introduces the Wasserstein Hypergraph Neural Network, which models nodes and hyperedge neighborhoods as distributions and aggregates them using Sliced Wasserstein Pooling. Unlike traditional mean or sum aggregators, this approach preserves geometric properties of distributions, capturing richer higher-order relationships. Experiments show that it significantly improves node classification performance on real-world hypergraph datasets.

**Strengths:**

1. Although the paper addresses a traditional GNN problem, it takes a very interesting perspective. Instead of using conventional aggregators like mean or sum, the authors represent the nodes within a hyperedge as a distribution. This approach better captures the internal structure and higher-order relationships of hyperedges.
2. The writing of the paper is very logical and well-structured. Although it involves some theoretical concepts, the explanations are clear and easy to follow.
3. This paper presents an innovative application of optimal transport theory in the context of GNNs.

**Weaknesses:**

The main drawback of the paper is the lack of complexity analysis and runtime comparisons. Since the optimal transport algorithm is generally not cheap computationally, this raises some concerns about the efficiency of the proposed method.

**Questions:**

1. This part feels vague in terms of its advantages—could it be presented in a more systematic way?

“For example, a hypergraph containing two clusters of nodes suggests a bimodal underlying distribution. On the other hand, a hyperedge where nodes are close in the feature space denotes a unimodal probability distribution, suggesting a homophilic behaviour. A hyperedge in which nodes have similar representations indicates a low-variance distribution, while a hyperedge with diverse nodes suggests a more uniform distribution (see Figure 1).”

2. Why was optimal transport chosen to address this problem? Could other methods for measuring distances between distributions not work as well?

3. Please include an analysis of time complexity and a comparison of the algorithm’s runtime efficiency.

P.S. While I am generally favorable toward this paper, as I find the idea of modeling hyperedges as distributions very appealing, I would be more cautious and might consider a weak accept if the authors cannot demonstrate acceptable computational efficiency.

---

> ### Author Response · Authors · 2025-11-18
>
> Thank you for your valuable review and positive feedback. We appreciate the time you’ve dedicated to evaluating our work and would like to respond to your concerns and questions.
>
> > This part feels vague in terms of its advantages—could it be presented in a more systematic way?
> “For example, a hypergraph containing two clusters of nodes suggests a bimodal underlying distribution. On the other hand, a hyperedge where nodes are close in the feature space denotes a unimodal probability distribution, suggesting a homophilic behaviour. A hyperedge in which nodes have similar representations indicates a low-variance distribution, while a hyperedge with diverse nodes suggests a more uniform distribution (see Figure 1).”
>
> We rephrased the sentence to focus more on the advantage of this view (namely, the fact that the geometry of the underlying distribution captures differences in group characteristics. Please let us know if this fixed the clarity issue.
>
>      “The shape of the hyperedge distributions can reveal different patterns in our data. For example, a hyperedge connecting nodes from two separate clusters forms a bimodal distribution, showing a mix of groups. A hyperedge where nodes are close together in feature space creates a unimodal distribution, reflecting similar, homophilic behavior. If the node features are very similar, the distribution has low variance, indicating a cohesive hyperedge. On the other hand, a hyperedge with more varied nodes produces a wider, more uniform distribution, showing diverse or loosely connected nodes (see Figure 1). This perspective allows us to interpret the internal structure of hyperedges in a more nuanced way, as the geometry of the distribution naturally reflects the characteristics of the node groups it connects.”
>
> > Why was optimal transport chosen to address this problem? Could other methods for measuring distances between distributions not work as well?
>
> We picked optimal transport since it already hinted at success in the set representation learning setup. However, other types of embeddings that are explicitly optimized to reflect distances or divergences between distributions could also offer valuable insights. Exploring these alternatives represent an interesting area for future work, representing a complementary view to our work.
>
> > Please include an analysis of time complexity and a comparison of the algorithm’s runtime efficiency
>
> A theoretical complexity analysis is provided in Appendix D, showing that the complexity of WHNN is influenced by both the number of reference points and the dimensionality of the neighbourhoods. In practice, there is a tradeoff between the complexity of the aggregation mechanism and the cost of training or inference. For current hypergraph benchmarks, this is typically not a limiting issue because hypergraphs are relatively small. If computational constraints become significant, the number of reference points used in Sliced Wasserstein Pooling (SWP) can be reduced to strike a more favourable balance between speed and accuracy. As demonstrated in the ablation studies (Table 5), varying the number of reference points has only a minor impact in our setting, and thus does not pose a major complexity concern. Nonetheless, hyperedge cardinality still exerts an independent influence.
>
> To provide a concrete comparison, we report runtime and memory usage for our best model alongside the DeepSet and PMA baselines on Citeseer (a medium-sized hypergraph) and DBLP_CA (a large hypergraph). While the Wasserstein aggregator introduces additional computational overhead compared to the simplest aggregation methods, the complexity of the aggregator (both V→E and E→V stages) remains comparable to that of the encoder.
> Inference time (in seconds) for the encoder (Enc) and aggregator (Agg) at each stage (V→E and E→V) measured on a Quadro RTX 8000 GPU:
>
> | Method  | Citeseer       |             |             |             | DBLP       |             |             |             |
> |---------|----------------|-------------|-------------|-------------|------------|-------------|-------------|-------------|
> |         | V-E Enc.       | V-E Agg.    | E-V Enc.    | E-V Agg.    | V-E Enc.   | V-E Agg.    | E-V Enc.    | E-V Agg.    |
> | WHNN    | 0.0022         | 0.0046      | 0.0025      | 0.0035      | 0.094      | 0.083       | 0.079       | 0.059       |
> | PMA     | 0.0025         | 0.0023      | 0.0030      | 0.0025      | 0.090      | 0.0089      | 0.071       | 0.0092      |
> | DeepSet | 0.0025         | 0.0010      | 0.0027      | 0.0012      | 0.091      | 0.0019      | 0.079       | 0.0019      |
>
> This table is also included in the appendix and is referenced in the experimental section of the main paper.

---

### Official Review · Reviewer_Mtxm · 2025-10-31

**Soundness:** 3
**Presentation:** 2
**Contribution:** 1
**Rating:** 2
**Confidence:** 3

**Summary:**

This paper introduces Wasserstein Hypergraph Neural Network (HNN). Whereas most existing HNNs use sum-based pooling to smooth information during message passing, the proposed method uses Wasserstein aggregation to preserve geometric information of node feature distribution within each hyperedge. The proposed method outperforms the baseline HNNs in several node classification benchmark datasets.

**Strengths:**

The key strengths are two-fold:
- The key idea is simple, intuitive, and solid. Existing set pooling functions may struggle to capture the full geometry of set-structured inputs, and using a more expressive pooling function to improve HNNs is a natural approach.
- The approach is also somewhat original to hypergraph learning. Intuitively, node feature distribution within each hyperedge could be distinct and informative for hypergraph learning.

**Weaknesses:**

I have five primary concerns:
- [limited novelty]: The key technical innovation is adapting Sliced Wasserstein Pooling (SWP), an existing module published in 2021, for HNNs. While the adaptation makes sense and seems to be performant, without a new technical innovation, the paper does not meet the standards of a top-tier conference.
- [thin content]: The experiments and analyses are thin. The authors used only 7 benchmark datasets with 8 (old) baseline HNNs. The method is evaluated only based on node classification performance. All the benchmark datasets have high label-homophily. No large dataset was tested. No formal or empirical analysis of the proposed method was provided. In fact, in a 9-page paper, the authors spent 3 pages solely on related work and background, which I do not think is generally acceptable.
- [weak method justification]: I do not agree with the authors' claim that the superior performance of the proposed method is due to 'capturing geometric information' of node features within each hyperedge. In most of the benchmark datasets used, the average hyperedge size is very small, e.g., 3-4 in Cora and Citeseer. Moreover, their input feature vectors are mostly sparse, multi-hot vectors. That is, I do not think such informative 'geometry' can be robustly identified in the benchmark datasets used. In fact, the authors did not provide any empirical analysis of the 'geometry' in the benchmark datasets. This drawback leaves the authors' key claim 'that this geometric information is highly relevant for hypergraph learning' (line 069) only speculative.
- [unfair hyperparameters]: The authors used self-loop addition as part of the tuned hyperparameters for each dataset. This is atypical and unfair. All the baseline methods, while they could, did not have the self-loop addition as their hyperparameter. This makes the empirical superiority of the proposed method suspicious.
- [copyright]: The baseline performance table is copied from the AllSet paper. This is okay, as long as the authors clearly mention that in the paper. However, I could not find it anywhere. This is a research ethics issue. Please revise properly.

I have other, less significant concerns:
- [missing details]: Details about the synthetic case in Figure 1 are missing, making it unclear whether the case is illustrative or based on formal analysis.
- [unclear writings]: Some writings are unclear. e.g., I do not understand 'This way, the hyperedges are not only characterised by the combination of their elements, but by the regions of the space where their elements are situated. The nodes became prototypes of the hyperedge behaviour.' Moreover, the authors write 'more sensitive to the geometric structure of the hyperedge' (line 228-229). However, if I understood correctly, they are not sensitive to the structure 'of hyperedge' but of 'node feature distribution within a hyperedge'. Please clarify.
- [unclear figures]: Some figures are quite confusing. E.g., in Fig. 2, what do node colors mean? What does the bar plot aim to depict? What do the contours aim to visualize? Adding legends, labels, and annotations may help.
- [minor errors]: Algorithm 1's line 4 is empty

Overall, in the current version, the paper is more like a workshop paper, with a simple, new, and promising idea and preliminary empirical outcomes.

**Questions:**

See weakness

---

> ### Author Response · Authors · 2025-11-18
>
> We sincerely appreciate your detailed review and valuable feedback on our paper. We would like to address your concerns and questions.
>
> > [limited novelty]: The key technical innovation is adapting Sliced Wasserstein Pooling (SWP), an existing module published in 2021, for HNNs. While the adaptation makes sense and seems to be performant, without a new technical innovation, the paper does not meet the standards of a top-tier conference.
>
> We acknowledge that Wasserstein-based embeddings have been explored in prior work over the past four years. However, as the reviewer also noted, our paper is the first to integrate Wasserstein pooling into hypergraph representation learning. We believe that this type of aggregator offers a promising new direction for improving message aggregation within hypergraph architectures. Historically, classical ideas such as Deep Sets, Set Transformers, convolutional and attentional mechanisms, and even message passing were initially developed for sets or graphs and only later adapted to hypergraphs, where they became influential components of the literature. In a similar spirit, our work takes an established concept and extends it into a domain where it has not yet been explored, thereby contributing to the field.
>
> > [thin content]: The experiments and analyses are thin. The authors used only 7 benchmark datasets with 8 (old) baseline HNNs. The method is evaluated only based on node classification performance. All the benchmark datasets have high label-homophily. No large dataset was tested. No formal or empirical analysis of the proposed method was provided. In fact, in a 9-page paper, the authors spent 3 pages solely on related work and background, which I do not think is generally acceptable.
>
> We would like to respectfully note that, although we reported results on seven datasets in the main paper, three additional datasets representing heterophilic settings are included in the appendix. All corresponding details can be found in Section A. While we are sorry that the reviewer had this impression, we believe that stating that ‘no empirical analyses of the model were provided’ may overlook the fact that the method was evaluated on ten benchmarks in total, with ablations evaluating the type of aggregator used, the type of encoder used, the kind of reference distribution and the number of reference points sampled from each distribution.
>
> Regarding the related work and background sections, our intention was to make the paper accessible to a broad audience, including readers without prior expertise in this specific area, an aspect that other reviewers explicitly appreciated.
>
> > [weak method justification]: I do not agree with the authors' claim that the superior performance of the proposed method is due to 'capturing geometric information' of node features within each hyperedge. In most of the benchmark datasets used, the average hyperedge size is very small, e.g., 3-4 in Cora and Citeseer. Moreover, their input feature vectors are mostly sparse, multi-hot vectors. That is, I do not think such informative 'geometry' can be robustly identified in the benchmark datasets used. In fact, the authors did not provide any empirical analysis of the 'geometry' in the benchmark datasets. This drawback leaves the authors' key claim 'that this geometric information is highly relevant for hypergraph learning' (line 069) only speculative
>
> We agree that current hypergraph datasets are somewhat restrictive and limited in diversity, which can make method comparisons challenging. However, some of the datasets we tested have relatively high cardinality—for example, House has a median hyperedge size of 34.8 with a maximum of 82, Senate has a median hyperedge size of 8 with a maximum of 99, and the median node degree in 20News is 537. By “geometry,” we refer to the distribution of node features in the latent space, which is not affected by one-hot input features and can remain informative even in small hyperedges. We would be happy to explore this further if the reviewer has concrete suggestions.
>
> > [missing details]: Details about the synthetic case in Figure 1 are missing, making it unclear whether the case is illustrative or based on formal analysis.
>
> In Figure 1 the three inputs were synthetically generated by sampling points from distribution with different configurations. The reported distances are empirically computed based on our code. Concretely, for the Wasserstein distance the reported number is computed as the L2 distance between the Wasserstein embeddings produced by our aggregator (which approximates the Wasserstein distance). We included all the details of generating the input and the reported metrics in the Appendix.

---

> ### Author Response · Authors · 2025-11-18
>
> > [unfair hyperparameters]: The authors used self-loop addition as part of the tuned hyperparameters for each dataset. This is atypical and unfair. All the baseline methods, while they could, did not have the self-loop addition as their hyperparameter. This makes the empirical superiority of the proposed method suspicious.
>
> We note that most of the hypergraphs that we compare against use self-loop by default. The reason why we included the use of self-loop as an additional hyperparameter is the particular probabilistic construction in our model. For the models using sum-based aggregators the self-loop acts basically similar to a residual connection. This is not the case for our model. In our particular construction the self loop corresponds to a probability distribution represented by a single point, which can be problematic in some scenarios, thus we decided to test with both options. We included this observation in the hyperaparameter section of the Appendix.
>
> > [copyright]: The baseline performance table is copied from the AllSet paper. This is okay, as long as the authors clearly mention that in the paper. However, I could not find it anywhere. This is a research ethics issue. Please revise properly.
>
> Our experimental pipeline is based on the code of EDHNN paper (which is also similar to AllSet). Thus the performance table is indeed the one from EDHNN. We are sorry for omitting this aspect. It is now included in the Experiment section of the main paper.
>
> > [unclear writings]: Some writings are unclear. e.g., I do not understand 'This way, the hyperedges are not only characterised by the combination of their elements, but by the regions of the space where their elements are situated. The nodes became prototypes of the hyperedge behaviour.' Moreover, the authors write 'more sensitive to the geometric structure of the hyperedge' (line 228-229). However, if I understood correctly, they are not sensitive to the structure 'of hyperedge' but of 'node feature distribution within a hyperedge'. Please clarify.
>
> We thank the reviewer for pointing out the unclear phrasing. When we say that “the hyperedges are not only characterised by the combination of their elements, but by the regions of the space where their elements are situated,” we mean that the hyperedge representation depends on the distribution of node embeddings (i.e., how these embeddings are positioned relative to one another). The expression “The nodes become prototypes of the hyperedge behaviour” refers to the fact that each node contributes as a representative point (or “prototype”) in the probability distribution we construct for the hyperedge. The aggregated representation therefore encodes information about both the participating nodes and their geometric configuration. Our use of “geometric structure of the hyperedge” was meant to describe this induced feature-space geometry, not a topology among the nodes themselves. To avoid confusion, we revised the text to state this explicitly: “designed to better reflect the geometric relationships among nodes features in the hyperedge”.
>
> > [unclear figures]: Some figures are quite confusing. E.g., in Fig. 2, what do node colors mean? What does the bar plot aim to depict? What do the contours aim to visualize? Adding legends, labels, and annotations may help.
>
> We apologise for the confusion. In the first column, the node colours carry no specific meaning, so we have removed them to improve clarity. In the middle and right columns, each color corresponds to a different reference point and is used to match the colours of the bars. The contours are used just to emphasize the probabilistic interpretation of the nodes in  the hyperedge, with one probability per hyperedge + an extra one for the reference distribution.
>
> In our algorithm, the points in the aggregated set and the reference points are sorted, and distances are computed between the corresponding sorted points. For example, the smallest point in the set is matched to the smallest reference point (colored red), while the largest point is matched to the largest reference point (purple). The resulting set of distances, represented by the heights of the bars, forms the final vectorial representation. We updated the caption of the figure to incorporate the explanations and we are working toward better highlighting this in the figure as well.

---

### Official Review · Reviewer_dN5k · 2025-10-31

**Soundness:** 3
**Presentation:** 3
**Contribution:** 3
**Rating:** 4
**Confidence:** 4

**Summary:**

Paper is interested in modeling neural networks for Hypergraphs: a generalization of graphs where any edge can connect zero-or-more nodes. Prior neural networks on hypergraphs follow message-passing paradigm, where each edge updates itself as an *average* of (a transformation of) its incident (endpoint) node features, then each node updates itself with an *average* of its incident hyperedges. This *average* (or sum) pooling captures only a simple statistic: the average. The paper pictorially shows why average is insufficient for representing a node. Suppose a graduate course with student average of 80, where most students receive a grade around the average. Suppose course with bimodal grades (centered at 90, and at 70). While both courses have a similar average, i.e. modeled the same with most (earlier) models, they should be represented differently. Paper rather tries to represent nodes "using distribution of edges" (and vice-versa: represent edges using distribution of nodes). Distance between 2 distributions is quantified using Wasserstein (W) distance. While W is intensive to compute for distributions over multiple dimensions, it is much easier to compute it for probability distributions over one (scalar) variable. (as sum of distances of inverse of CDF), allowing one to approximate the distance between 2 distributions (over higher dimension) by taking many random direction lines, and projecting the distributions upon these lines i.e. break-down the problem into many 1D W-distances, and average them, to approximate the distance in higher dimensions. Further, rather than calculating the approximate distance (even in the 1D case), paper mentions that they can project onto a latent space where euclidean-distances are trained to approximate W-distances.

Sorry for the long description :)

All-in-all -- I like the paper a lot. However, there are two main things that need to be addressed for acceptance consideration (IMO):
1- Clarity around the math / algorithm (see Weaknesses).
2- Novelty. There are Wasserstein GNNs (https://arxiv.org/abs/2102.03450)

**Strengths:**

* Paper is well-written. I learned a bunch, as perhaps apparent from the above summary. It provides both good mathematical understanding (step-by-step walkthrough), as well as demonstration figures in main paper and Appendix.
* The paper utilizes different grounds-up concepts, which is "refreshing" in LLM era.
* Good experimental results

**Weaknesses:**

## main weakness
The main weakness is in the clarity of the method. Not in terms of the write-up/motivation/high-level description (I think those are clear), but in terms of actually implementing or replicating this method. The math/algorithm section has loose ends, and personally, while I appreciate this work, I am unable to implement it from reading this paper alone. Great papers, on the other hand, I am able to implement them by reading only those papers. Let's try to improve this paper from "Good" to "Great" in order to have it appropriate for ICLR.

Specifically,

* what is the "interpolation" mentioned in Step 2 (Page 7) or line 12 in Alg.2. It seems that it will discard entries if there are too many, or invent new entries if there are only a few of them. I would guess that inventing could imply taking average of existing points?

* Line 5 of Alg2: Does it expand the dimensionality of X?
* Line 7 of Alg2: what are the vectors (reference points) sorted by? by their CDF value?
* Line8 onwards of Alg2: what is the lowercase symbol $s$? Of course, upper-case $S$ is the set of neighbors -- does lower-case $s$ correspond to the center (node or edge) for which its neighbros are recorded in upper-case $S$? If so, perhaps consider looping $(s, S) \in \mathcal{N}.items()$ -- or something similar. We are lucky that most ML folks are able to read pythonic-pseudocode.
* Line 18 of Alg2: is $W$ vector of scalars? In general, **please add sizes of all tensors**. Does this line sum-over a dimension? (weighted sum, with weights present in vector $W$)
* Alg1 line 7 or Step 1 on page 7 -- what is the reference distribution $q$? does it contain some anchor (randomly-sampled) nodes/edges? Do they have to go through the encoder (Line 11)?


## Novelty

Have you seen: https://arxiv.org/pdf/2102.03450 ? In what way does your method differ? Is it only in the application (Hypergraphs VS graphs)? or is the entire construction different?


## minor weakness:
* Typo: Perhaps replace "inner integral" with "integral" on line 201? I get that this becomes "inner" once plugged-into Eq.2 but perhaps the text is confusing as word "inner" shows up before Eq.2

**Questions:**

* Are the nodes of the hyper-edges ordered (i.e. order of nodes matter, as a generalization of directed graphs)? The math does not seem ordered. But what if the data is ordered? How can one extend the model to handle this data? E.g., (<Student> studied <program> at <university>)
* Please answer the questions in weaknesses, importantly, by updating the paper.

---

> ### Author Response · Authors · 2025-11-18
>
> We thank the reviewer for pointing out the aspects in which the current paper can improve clarity. This is much appreciated.
>
> >*what is the "interpolation" mentioned in Step 2 (Page 7) or line 12 in Alg.2. It seems that it will discard entries if there are too many, or invent new entries if there are only a few of them. I would guess that inventing could imply taking average of existing points?*
>
> Indeed the general purpose of interpolation is to “drop” or “invent” points when the cardinality is not matching the desired one. In our experiments, we apply simple linear interpolation to samples from 1-dimensional distributions (the slices). More precisely, for a hyperedge containing K sample points, we treat these points as evaluations of the inverse CDF at K uniformly spaced quantiles in [0,1]. To obtain a representation with R points instead, we evaluate the inverse CDF at R  new uniformly spaced quantiles by linearly interpolating between the original K values. This produces an R-point discretization that is consistent with the distribution implied by the original samples. During the development we also experimented with other interpolation options (as supported by https://docs.scipy.org/doc/scipy/reference/generated/scipy.interpolate.interp1d.html) including the nearest neighbour one which just copy or drop points depending on the need, but we haven’t noticed substantial differences in performance. We included this explanation in the appendix C to ensure easier reproduction.
>
> >Line 5 of Alg2: Does it expand the dimensionality of X?
>
> In our experiments we set the number of slices the same as the number of  hidden dimensions so the dimensionality is not changed here ( $N \times L \rightarrow N \times L$ ). The purpose of this step is only to match the theoretical interpretation of the algorithm. To accurately approximate the high-dimensional distribution the original features are projected into several one-dimensional ones using random directions on the unit sphere. Concretely, in our case, we use learnable directions ($\theta$) restricted to have magnitude one. We included the dimensions in the algorithm to avoid confusions.
>
> >Line 7 of Alg2: what are the vectors (reference points) sorted by? by their CDF value?
>
> The reference points are, independently  for each slice, sorted by their value. We included that comment in the algorithm.
>
> > Line8 onwards of Alg2: what is the lowercase symbol ? Of course, upper-case  is the set of neighbors -- does lower-case  correspond to the center (node or edge) for which its neighbros are recorded in upper-case ? If so, perhaps consider looping  -- or something similar. We are lucky that most ML folks are able to read pythonic-pseudocode.
>
> Yes, it is precisely the node for which the neighbourhood is computed. We are sorry for the confusion. The intention was to keep it as an upper-case letter with X_S representing the  embedding of the set of elements  represented as “S”. But moving to edge/node index might be indeed more clear. We made the change in the algorithm and we hope it fixed the confusion.
>
> >Line 18 of Alg2: is  vector of scalars? In general, please add sizes of all tensors. Does this line sum-over a dimension? (weighted sum, with weights present in vector )
>
> We totally agree that adding sizes makes the algorithm much easier to understand. Thus, we included all the sizes now. The purpose of the equation in Line 18 is to reduce the dimension of the output from the R \times L dimension produced by the Wasserstein embedding to an L-dimensional vector as required by the future stages. Any such operation can be applied. To maintain the number of parameters low, in our implementation we perform an elementwise multiplication with a learnable R \times L  matrix followed by a reduction on the R-dimension.
>
> > Alg1 line 7 or Step 1 on page 7 -- what is the reference distribution ? does it contain some anchor (randomly-sampled) nodes/edges? Do they have to go through the encoder (Line 11)?
>
> The reference distribution indeed represents the set of anchors randomly sampled. They do not need to go through the encoder since they are directly sampled to represent the anchors in the embedded space. However, since they are only anchors used as the space origin, except for some trivial cases (e.g. picking all of them in the same point) the way we chose/encode them should not have a big impact on the results.

---

> > ### Author Response · Authors · 2025-11-18
> >
> > > Have you seen: https://arxiv.org/pdf/2102.03450 ? In what way does your method differ? Is it only in the application (Hypergraphs VS graphs)? or is the entire construction different?
> >
> > Thank you for the reference which to be honest we were not aware of before. The motivation in that paper is similar to ours, but the  construction is different. Our Wasserstein aggregator uses Wasserstein embedding to create a representation such that the distance is preserved while their method uses Wasserstein barycenters as a way to capture node uncertainty. More specifically:
> >
> > * Barycenter vs Wasserstein embedding: For a set of distributions, **Wasserstein barycenter** generates a **distribution** such that it is the closest in Wasserstein distance to all the distributions in the aggregated set. On the other hand, for a set of points sampled from a single distribution, **Wasserstein embedding** generates a **vectorial representation** such that, given two distributions, the Wasserstein distance between them can be approximated using the computed representations. In other words, Wasserstein barycenter operates on a set of distributions while Wasserstein embeddings operates on each distribution (or on samples from a distribution) individually.
> >
> > * Node as distribution vs neighbourhoods as distribution: The mentioned paper looks at **each node as a different distribution** (characterised by the observed features + uncertainty given by missing attributes) and uses Wasserstein barycenter to compute an average of these distributions. Instead we look at **nodes as observed points of the neighbourhood distribution** and use Wasserstein embedding to create a representation of this neighbourhood.
> >
> > Since the paper is very relevant for the intersection between Wasserstein literature and graph representation learning, we included it in the related work section.
> >
> > > Typo: Perhaps replace "inner integral" with "integral" on line 201? I get that this becomes "inner" once plugged-into Eq.2 but perhaps the text is confusing as word "inner" shows up before Eq.2
> >
> > Thank you for pointing this out to us. We replaced the "inner integral" with "integral".
> >
> > > Are the nodes of the hyper-edges ordered (i.e. order of nodes matter, as a generalization of directed graphs)? The math does not seem ordered. But what if the data is ordered? How can one extend the model to handle this data? E.g., ([object Object] studied [object Object] at [object Object])
> >
> > Indeed, the current formulation is designed for undirected hypergraphs and therefore assumes no inherent ordering in the data. A particular class of directed hypergraphs involves hyperedges in which a subset of nodes are designated as in-flow nodes and the remaining ones as out-flow nodes (e.g., reactants and products in chemical reactions). For such data, the message-passing procedure can be adapted so that, in the node-to-edge stage, the hyperedge representation is constructed by aggregating only the representations of the in-flow nodes. Conversely, in the edge-to-node stage, only the out-flow nodes are updated based on the aggregated hyperedge representation. This modification allows the algorithm to follow the natural flow of information implied by the input and output nodes. However, if a strict ordering of all elements within each hyperedge is required (as in the reviewer’s example), we recommend incorporating positional encodings similar to those used in transformer architectures.

---

> > > ### Comment · Reviewer_dN5k · 2025-11-20
> > >
> > > Thank you for sharing how you would use this method for "ordered nodes", and for updating the algorithm (using index $i$ to loop over zip(nodes, neighbors) and similarly for edges).
> > >
> > > I do think that this is a good ICLR contribution and I could see myself using it for a few of my applications. Therefore, I will vote to accept this paper.

---

### Meta-Review · Area_Chair_PTiB · 2026-01-03

**Summary:**

The reviewers have raised the following major concerns:

(1) Clarity of the method.

(2) Novelty.

(3) Regarding the datasets, it is unclear whether the features contain sufficient geometric information to support the proposed method.

(4) Lack of complexity analysis and comparison of runtime.

(5) Theoretical Grounding not thoroughly discussed.

(6) Incremental performance gain.

**Reviewer Concerns:**

The revision has addressed several concerns raised by the reviewers (e.g., points (1) and (4)). Nevertheless, some important issues remain.

I share Reviewer Mtxm’s assessment regarding the novelty of the work. SWP has been studied and developed for several years, and while its incorporation into HNNs may be practically useful, it does not appear to constitute a sufficiently novel technical contribution for ICLR.

In addition, the proposed model lacks strong theoretical support, as noted by Reviewer qh5o and acknowledged by the authors themselves. At the same time, the empirical performance gains are relatively incremental, which further weakens the case for the necessity of SWP in the considered graph and hypergraph learning tasks.

More broadly, based on this submission and my related experience, there is currently no clear evidence that HNNs consistently outperform standard GNNs on commonly used graph benchmarks. Consequently, I am not convinced that these datasets are the most appropriate testbeds for evaluating HNN-specific design choices, a concern also raised by Reviewer Mtxm. It may be more compelling for future work to introduce novel datasets that explicitly and conceptually emphasize higher-order relationships, thereby better motivating architectural elements designed specifically for hypergraphs.

**Reviewer Scores:**

Reviewer dN5k has indicated his/her willingness to increase the score. The other reviewers are likely to maintain their original scores.

---

### Decision · Program_Chairs · 2026-01-26

Reject